# Type-2 CD8+ T-cell formation relies on interleukin-33 and is linked to asthma exacerbations

Esmee K. van der Ploeg[1,2], Lisette Krabbendam [1,6], Heleen Vroman[1,6], Menno van Nimwegen[1], Marjolein J. W. de Bruijn[1], Geertje M. de Boer [1,3], Ingrid M. Bergen[1], Mirjam Kool [1], Gerdien A. Tramper-Standers [4,5], Gert-Jan Braunstahl [1,3], Danny Huylebroeck [2], Rudi W. Hendriks [1,7] & Ralph Stadhouders [1,2,7] ✉

CD4+ T helper 2 (Th2) cells and group 2 innate lymphoid cells are considered the main producers of type-2 cytokines that fuel chronic airway inflammation in allergic asthma. However, CD8+ cytotoxic T (Tc) cells - critical for anti-viral defense - can also produce type-2 cytokines (referred to as 'Tc2' cells). The role of Tc cells in asthma and virus-induced disease exacerbations remains poorly understood, including which micro-environmental signals and cell types promote Tc2 cell formation. Here we show increased circulating Tc2 cell abundance in severe asthma patients, reaching peak levels during exacerbations and likely emerging from canonical IFNγ+ Tc cells through plasticity. Tc2 cell abundance is associated with increased disease burden, higher exacerbations rates and steroid insensitivity. Mouse models of asthma recapitulate the human disease by showing extensive type-2 skewing of lung Tc cells, which is controlled by conventional type-1 dendritic cells and IFNγ. Importantly, we demonstrate that the alarmin interleukin-33 (IL-33) critically promotes type-2 cytokine production by lung Tc cells in experimental allergic airway inflammation. Our data identify Tc cells as major producers of type-2 cytokines in severe asthma and during exacerbations that are remarkably sensitive to alterations in their inflammatory tissue micro-environment, with IL-33 emerging as an important regulator of Tc2 formation.

Asthma is a heterogeneous chronic inflammatory disease of the airways involving a complex interplay of various immune and lung structural cells that causes bronchial hyper-reactivity, excessive mucus production, and airway narrowing[1]. In eosinophilic asthma, allergens damage or activate epithelial cells to release inflammatory 'alarmin' cytokines thymic stromal lymphopoietin (TSLP), interleukin-25 (IL-25), and IL-33, which directly activate a range of immune cells including dendritic cells (DC), lung-resident group 2 innate lymphoid cells (ILC2) and CD4+ T helper (Th) cells[2]. Activated DCs take up the allergens and migrate to the lymph nodes to present the antigens to naive Th cells, promoting the differentiation of type-2 T helper (Th2) cells that subsequently migrate back to the lung. Both activated Th2 cells and ILC2s

[1]Department of Pulmonary Medicine, Erasmus MC, University Medical Center, Rotterdam, The Netherlands. [2]Department of Cell Biology, Erasmus MC, University Medical Center, Rotterdam, The Netherlands. [3]Department of Respiratory Medicine, Franciscus Gasthuis and Vlietland, Rotterdam, The Netherlands. [4]Department of Pediatric Medicine, Franciscus Gasthuis and Vlietland, Rotterdam, The Netherlands. [5]Department of Neonatology, Sophia Children's Hospital, Erasmus MC, University Medical Center, Rotterdam, The Netherlands. [6]These authors contributed equally: Lisette Krabbendam, Heleen Vroman. [7]These authors jointly supervised this work: Rudi W. Hendriks, Ralph Stadhouders. ✉e-mail: r.stadhouders@erasmusmc.nl

produce large amounts of the type-2 cytokines IL-4, IL-5, IL-9, and IL-13, which in turn trigger hallmark asthma symptoms[2,3]. Given their central role in asthma pathogenesis, type-2 cytokines, and Th2 cells are major targets for therapy[4]. This includes treatment with corticosteroids to which the canonical adaptive type-2 immune response appears highly susceptible[5,6]. Nevertheless, many patients with eosinophilic asthma fail to achieve proper symptom control using steroids, suggesting that alternative and possibly more steroid-resistant type-2 cytokine producers may critically contribute to therapy-resistant asthma. For example, we recently showed that CD45RO[+] inflammatory ILC2s are steroid-resistant and elevated in the circulation of patients with uncontrolled asthma[7].

Asthma exacerbations are often associated with respiratory viral infections[1], which trigger a potent CD8[+] cytotoxic T (Tc) cell response[8]. Canonically, activated Tc cells produce type-1 inflammatory cytokines such as IFNγ and release cytotoxic molecules (e.g., granzymes, perforins). The resulting cytokine milieu can suppress type-2 inflammation[9,10], leading to a seemingly paradoxical relationship between respiratory viral infections and asthma exacerbations. However, Tc cells can also respond to asthma-associated type-2 activation signals, including IL-33 and IL-4[11]. Under inflammatory conditions, Tc cells can be phenotypically skewed from IFNγ production (a Tc1 cell) towards the synthesis of type-2 cytokines (Tc2) or IL-17 (Tc17)[12–14]. Such Tc cell plasticity can thus generate non-canonical type-2 cytokine-producing Tc cells, potentially contributing to asthma exacerbations and therapy-resistance given their reduced sensitivity to steroids[15]. Indeed, studies have detected Tc2 cells in blood and lung samples of asthma patients; their numbers often correlating with long-term decline of lung function[16,17], poor symptom control[18–22], or death[23]. Importantly, studies using mouse models of allergic airway inflammation (AAI) have shown that antibody-mediated depletion or genetic deletion of CD8[+] T cells can reduce airway hyperresponsiveness and eosinophilia[14,24–28].

Despite the relevance of Tc cell biology for type-2 immunity and linked immunopathologies in humans, many aspects remain poorly understood, including: (1) the precise functional capacities of these cells, (2) their association with hallmarks of severe asthma such as exacerbations, and 3) the micro-environmental signals that drive their formation.

Here, we address these issues using a combination of in-depth profiling of Tc/Th cells from stable or exacerbating asthma patients and analyses of transgenic mice subjected to eosinophilic airway inflammation. These analyses reveal phenotypic skewing of the Tc cell compartment towards type-2 cytokine production, which associates with increased asthma severity and reaches peak levels during exacerbations. In vivo, Tc2 cell generation is promoted by IL-33 and inhibited by IFNγ or type-1 conventional DC (cDC1) activity. Together, our results support an important role in the functional adaptation of Tc cells by alarmins in severe asthma.

## Results

### Increased type-2 cytokine production by circulating Tc cells in asthma patients

To compare the functional phenotype of the Tc and Th cell compartment in asthma, we measured the cytokine profile of peripheral blood (PB) T cells in a cohort of 55 clinically well-characterized asthma patients and 17 healthy controls (HC; Table S1) using flow cytometry (gating strategy in Supplementary Fig. 1A). Whereas total Tc/Th cell frequencies were similar between asthma patients and HCs (Supplementary Fig. 1B), production of the type-2 cytokines IL-5, IL-9, and IL-13 by both Tc and Th cells was significantly higher in asthma patients than in HCs (Fig. 1A). By contrast, IFNγ, IL-4, and IL-17A production did not differ (Supplementary Fig. 1C, D). Interestingly, we observed elevated frequencies of IFNγ[+]IL-5[+], IFNγ[+]IL-9[+], and IFNγ[+]IL-13[+] double-producing Tc cells, supporting type-2 skewing of IFNγ[+] Tc1 cells in asthma (Fig. 1B,

Supplementary Fig. 1E). Similar results were obtained for Th cells (Fig. 1A, Supplementary Fig. 1D, F). Previous work indicated that Tc2 cells express the prostaglandin D2 receptor CRTH2[29,30], which is in line with the increased abundance of CRTH2[+]CD4[-] T cells we detected in a different asthma patient cohort (Supplementary Fig. 2A). To rule out that type-2 cytokine production by Tc cells was merely a consequence of PMA/ionomycin stimulation, we induced T-cell receptor (TCR) activation in cultured Tc cells using anti-CD3/CD28 beads. Also in the absence of PMA/ionomycin restimulation, Tc cells produced similar levels of type-2 cytokines, which were highest in the CRTH2[+] fraction (Supplementary Fig. 2B). These data demonstrate that Tc2 cells produce type-2 cytokines when stimulated under physiologically relevant conditions.

Correlation analysis revealed coordinated production of individual cytokines within the Th and Tc cell compartments of asthma patients (Fig. 1C). Strong positive correlations existed between IL-5 and IL-9 levels, regardless of T-cell origin, whereas correlations between IFNγ/IL-4/IL-13 were much stronger in Th than Tc cells. Across Th and Tc cell compartments, strong correlations existed among the individual cytokines, except IL-17A (Fig. 1D). Strikingly, IL-5 and IL-9 levels showed particularly strong associations ($R > 0.7$, $P < 0.001$) both within and between Th and Tc cells (Fig. 1C, D). Although similar positive correlations were observed in HCs, they displayed several negative correlations between cytokine-positive Th and Tc cells (e.g., IL-4[+] Th and IL-5[+]/IL-13[+] Tc cells; Supplementary Fig. 2C, D), indicating reduced phenotypic separation across blood T-cell compartments in asthma patients compared to HCs. Unbiased clustering using tSNE revealed clusters of IL-4, IL-5, IL-9, and IL-13-producing cells separated from a canonical IFNγ[+] Tc1 cell cluster (Fig. 1E). In line with our correlation analysis, a subset of IL-4[+] Tc cells co-localized with IL-13[+] or IFNγ[+] Tc cells, whereas IL-5[+] and IL-9[+] Tc cells localized to largely distinct yet adjacent clusters. Analysis of cytokine-producing Th cells yielded different patterns, although subset co-localization was largely similar (Supplementary Fig. 2E).

Together, these data show that circulating Th and Tc cells in asthma patients exhibit increased type-2 cytokine production capacity compared to HCs. Increased frequencies of Tc cells producing both IFNγ and type-2 cytokines in patients suggest that asthma coincides with elevated Tc1-to-Tc2 cell skewing, which occurs in a concerted manner across both T-cell compartments.

### Type-2 skewing of Tc and Th cells is linked to asthma disease severity

We next sought to link type-2 skewing of Tc cells to specific asthma disease characteristics. As no differences in type-2 cytokine production were observed between atopic and non-atopic, or between T2 and non-T2 asthma patients (see Methods; Table S2), we focused on symptom severity. Asthma control questionnaire (ACQ, a standardized measure of disease symptom control) scores were used to subdivide patients into 'controlled' (ACQ < 0.75), 'partially controlled' (ACQ = 0.75-1.5), and 'uncontrolled' (ACQ > 1.5) categories[31]. Although IL-5[+] and IL-13[+] Tc cell levels were increased in all patient groups compared to HCs, IL-9[+] Tc cell frequencies were specifically associated with uncontrolled asthma (Supplementary Fig. 3A). IL-4, IFNγ, and IL-17A production was not significantly increased in any subgroup (Supplementary Fig. 3A); similar results were obtained for Th cells. Importantly, Tc cells producing type-2 cytokines together with either IFNγ or IL-17, indicative of phenotypic skewing, were particularly associated with poor disease control (Supplementary Fig. 3B).

We next grouped asthma patients based on their exacerbation frequencies in the year prior to inclusion, which is a key indicator of disease severity[32]. Compared to HCs, increased frequencies of IL-5[+], IL-9[+], and IL-13[+] Tc cells were only observed in patients with ≥2 exacerbations, as patients with 0-1 exacerbation showed no increase (Fig. 2A).

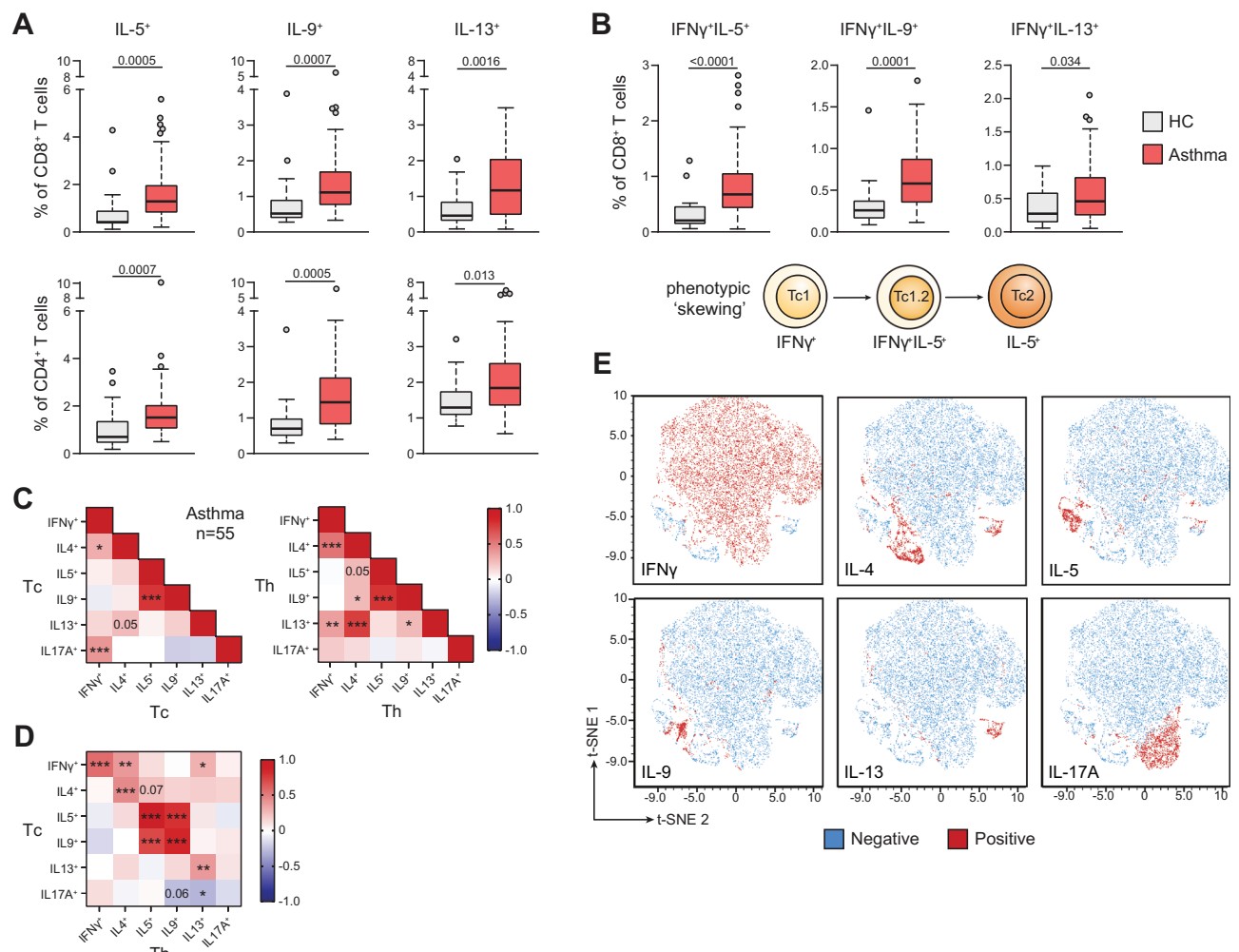

**Fig. 1 | Type-2 skewing of circulating Tc cells in asthma. A** Quantification of IL-5⁺, IL-9⁺, and IL-13⁺ Tc and Th cells using flow cytometry in PB samples of 17 HC individuals and 55 asthma patients. **B** Quantification of IFNγ⁺IL-5⁺, IFNγ⁺IL-9⁺, and IFNγ⁺IL-13⁺ double-producing Tc cells in PB samples of HC individuals and asthma patients. **C**, **D** Correlation matrices of cytokine production between (**C**) and across (**D**) Th/Tc cell compartments. **E** tSNE analysis of flow cytometry data using all cytokine-producing Tc cells from asthma patients, depicting either IFNγ⁺, IL-4⁺, IL-5⁺, IL-9⁺, IL-13⁺, or IL-17A⁺ producing Tc cells in red. Symbols in **A**, **B** represent individual donors; bars indicate mean values ±SEM. *P < 0.05, **P < 0.01, ***P < 0.001, ****P < 0.0001 (two-tailed Mann–Whitney U test, Pearson correlation coefficient analysis). PB peripheral blood, HC healthy control. Source data are provided as a Source Data file.

IL-4 and IFNγ levels did not differ between subgroups, and IL-17A⁺ Tc cell frequencies were lowest in patients with ≥2 exacerbations (Supplementary Fig. 4A). Similar results were obtained for IL-5⁺ and IL-9⁺ Th cells (Supplementary Fig. 4B). Levels of Tc cells that expressed IL-5, IL-9 or IL-13 together with IFNγ or IL-17 were specifically increased in patients with ≥2 exacerbations (Fig. 2B, Supplementary Fig. 4C).

Poor symptom control and frequent exacerbations are associated with insensitivity to inhaled corticosteroids. To investigate the relationship between steroid dose and T-cell cytokine production, we compared patients with a low/medium or high steroid dose (<1000 versus >1000 steroid bioequivalent units). High steroid intake was associated with elevated IL-5⁺, IL-9⁺, and IL-13⁺ Tc cell frequencies when compared with patients receiving a lower dose (Fig. 2C). IL-4⁺, IFNγ⁺ and IL-17A⁺ Tc cell levels were not different (Supplementary Fig. 5A). For Th cells, only IL-17A-producing cells were increased in patients with high steroid dose intake (Supplementary Fig. 5B). Interestingly, IL-9⁺ Tc cell abundance positively correlated with frequencies of CD45RO⁺ ILC2 (Supplementary Fig. 5C), a recently identified subset of inflammatory ILC2s associated with steroid-resistance and severe asthma[7].

In summary, these results show that increased Tc2 cell abundance, and possibly type-2 skewing of Tc1 cells, are linked to uncontrolled asthma, frequent exacerbations, and high steroid intake.

## A major shift in circulating Tc1/Tc2 balance occurs during asthma exacerbations

Disease exacerbations induced by respiratory viruses are a major cause of morbidity for asthma patients. As the canonical function of Tc cells is to provide antiviral immunity, we investigated Tc cell compartment dynamics as patients progress from stable disease to an exacerbation. Upon exacerbating, asthma patients showed significantly decreased frequencies of IFNγ⁺ and simultaneously increased frequencies of IL-4⁺, IL-5⁺, and IL-9⁺ Tc and Th cells (Fig. 2D, Supplementary Fig. 5D-E). Notably, Tc2 cell abundance was very comparable to frequencies of Th2 cells in HCs, stable asthma patients and during an exacerbation (Fig. 2E). Depending on the specific cytokine and sample group, intracellular cytokine levels on a per cell basis differed between Tc2 and Th2 cells (Supplementary Fig. 5F). Proportionally, Tc2 cells made up ~11 and ~14% of the total circulating cytokine-producing Tc cells in HCs and stable asthmatics, respectively, whereas this percentage increased to ~23% during an active exacerbation (Fig. 2F)−a larger relative increase than seen for Th2 cells (Supplementary Fig. 5G). Correlation analysis revealed that concerted cytokine expression profiles observed among Tc cells were further strengthened during exacerbations, whereas Th2 cells showed loss of several positive correlations

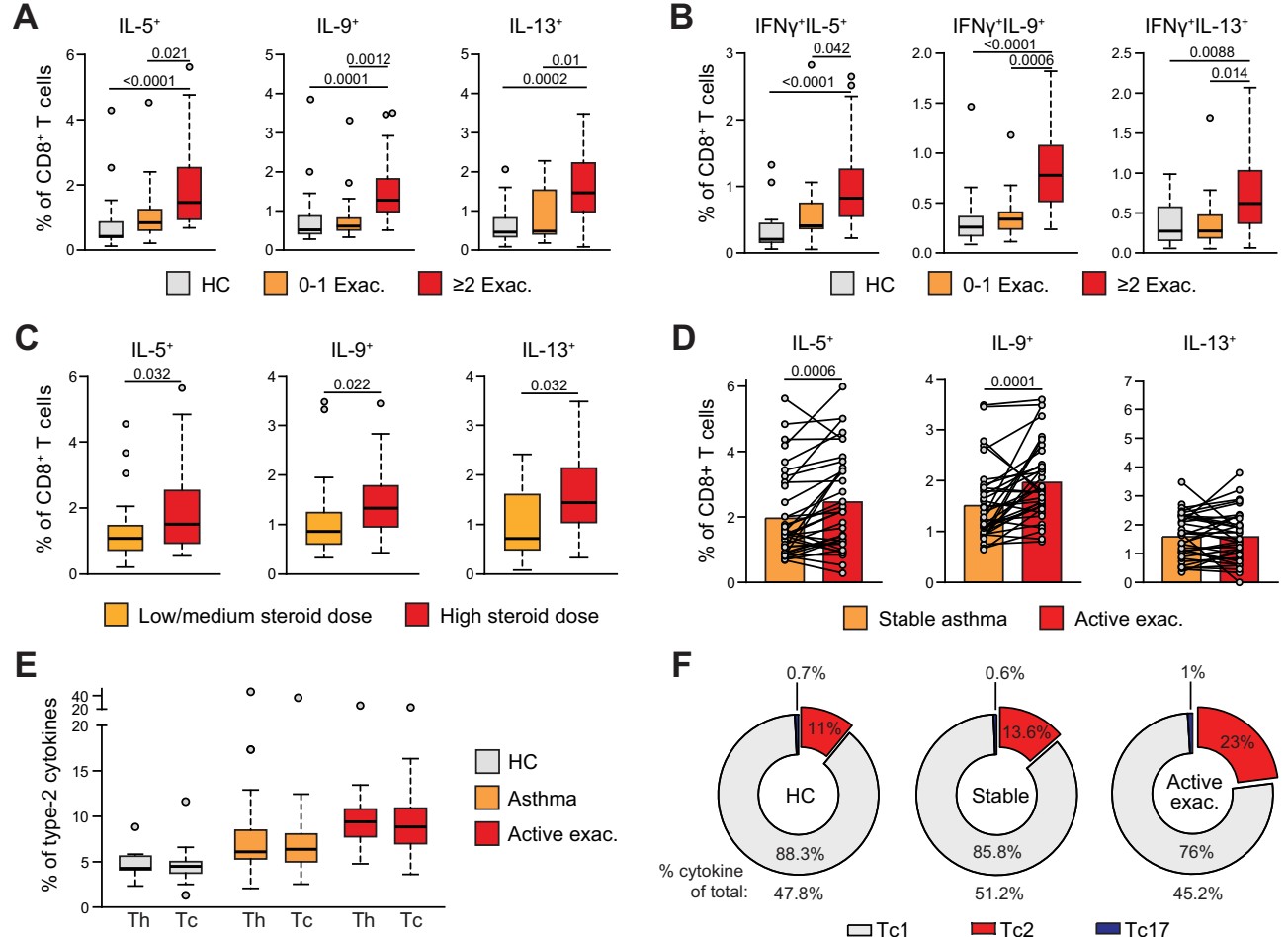

**Fig. 2 | Type-2 skewing of Tc cells is linked to asthma symptom severity.**
**A** Quantification of IL-5+, IL-9+, and IL-13+ Tc cells using flow cytometry in PB samples of 17 HC individuals and asthma patients with 0-1 exacerbations ('Exac.', $n = 16$) or ≥2 exacerbations in the previous year ($n = 38$). **B** Quantification of IFNγ+IL-5+, IFNγ+IL-9+, and IFNγ+IL-13+ double-producing Tc cells in PB samples of 17 HC individuals and asthma patients with 0-1 exacerbations ('Exac.', $n = 16$) or ≥2 exacerbations in the previous year ($n = 38$). **C** Quantification of IL-5+, IL-9+, and IL-13+ Tc cells in PB samples of asthma patients with low/medium steroid dose intake (<1000 bioequivalent units, $n = 23$) and high steroid dose intake (>1000 dose bioequivalent units, $n = 30$). **D** Quantification of IL-5+, IL-9+, and IL-13+ Tc cells in 35 paired PB samples of asthma patients during stable disease and during an active exacerbation ('Active exac'). **E** Quantification of type-2 cytokine production (combining all IL-4, IL-5, IL-9, and IL-13 producing cells) by Th and Tc cells in 17 HC individuals, 55

asthma patients, and during an active exacerbation ($n = 35$). **F** Pie charts summarizing proportions of Tc1, Tc2, and Tc17 cells in HCs, asthma patients, and during an active exacerbation. '% cytokines of total:' indicates percentage of all Tc cells that produce one of the cytokine assayed. Symbols in **A**–**E** represent individual donors; Bars indicate mean values ±SEM. *$P < 0.05$, **$P < 0.01$, ***$P < 0.001$, ****$P < 0.0001$ (two-tailed Mann–Whitney $U$ test, Wilcoxon rank-sum test, Kruskal–Wallis test corrected for multiple testing or Pearson correlation coefficient analysis), PB peripheral blood, HC healthy control, Tc1: cells producing IFNγ excluding double-producers of type-2 or type-3 cytokines, Tc2: any cell producing either IL-4 IL-5 IL-9 or IL-13, Tc17: cells producing IL-17A excluding double-producers of type-1 or type-2 cytokines. PB peripheral blood, HC healthy control. Source data are provided as a Source Data file.

(Supplementary Fig. 5H). Across stable and exacerbating states, we observed strong positive correlations in cytokine production within the Tc population that were much less pronounced in Th cells (Supplementary Fig. 5I) indicating tighter coordination of cytokine production amongst Tc cells in severe asthma.

Taken together, our analyses reveal an elevated presence of circulating Tc2 cells during asthma exacerbations, comprising a substantial portion of total cytokine-producing Tc cells. Importantly, quantitative increases in Tc2 cell frequencies rival those seen for Th2 cells.

## Tc2 cells can induce eosinophilic AAI in vivo
To demonstrate that Tc2 cells are capable of inducing eosinophilic airway inflammation in vivo, we first exposed naïve Tc (OTI) or Th (OTII) cells carrying a transgenic TCR specific for the ovalbumin (OVA) model allergen to anti-CD3/CD28 beads and IL-4 to induce Tc2/Th2 polarization in vitro. As previously reported[33], these

conditions induced potent type-2 cytokine production in Tc cells, with a concomitant loss of canonical Tc1 markers such as IFNγ and T-bet (Fig. 3A, Supplementary Fig. 6A–C). Adoptive transfer of Tc1, Tc2, and Th2 cells into wildtype (WT) naïve recipient mice was followed by 4 days of intratracheal exposure to OVA and house dust mite (HDM) treatment (Fig. 3B). Analysis of bronchoalveolar lavage (BAL) fluid showed that Tc2 cell transfer and activation in vivo induced a marked infiltration of GR1+ inflammatory eosinophils in the airways, which was similar to the nature and magnitude of inflammation observed with Th2 cells (Fig. 3C, Supplementary Fig. 6D). In contrast, stimulated Tc1 cells or Tc2 cells not receiving in vivo TCR stimulation did not trigger eosinophilic airway inflammation. In line with the induction of eosinophilia, transferred Tc2 cells exhibited robust IL-5 production upon activation in vivo (Fig. 3D, Supplementary Fig. 6E)[34].

In conclusion, these results strongly support a bona fide role of Tc2 cells as drivers of eosinophilic airway inflammation.

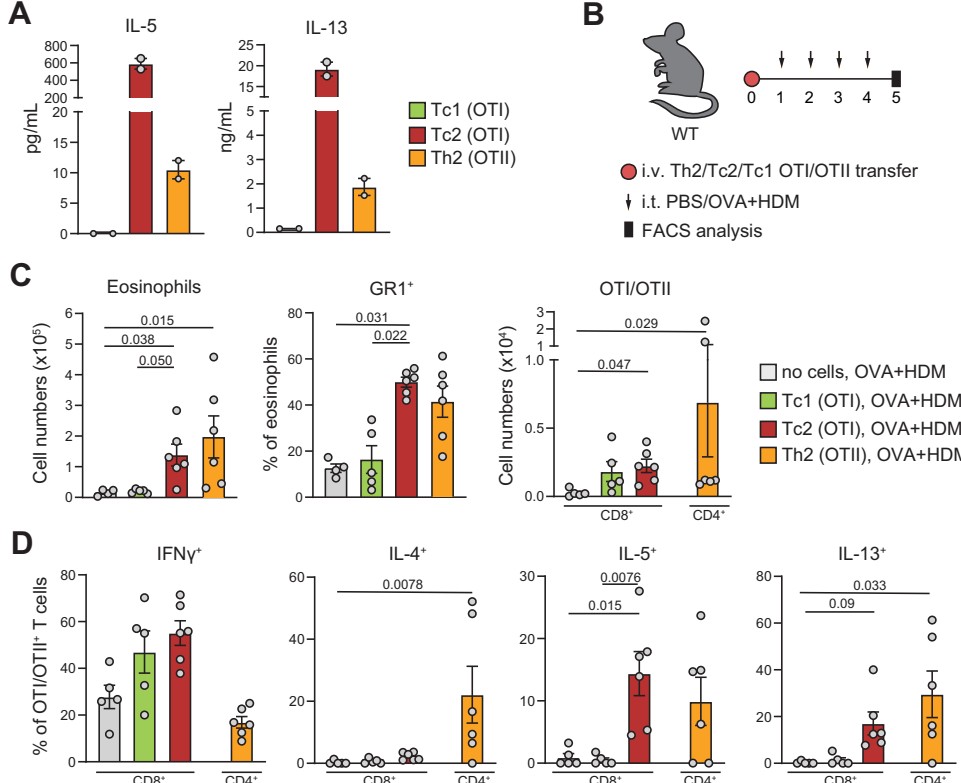

**Fig. 3 | Tc2 cells can induce eosinophilic airway inflammation in vivo. A** IL-5 and IL-13 levels were measured by ELISA in culture supernatants of the indicated cells, derived from two OTI or OTII mice. naive OTI or OTII cells were cultured with anti-CD3, anti-CD28 in the presence of IL-2 for Tc1 polarization, or IL-2, IL-4, and anti-IFNγ for Tc2 polarization. For Th2 polarization, cells were cultured in the presence of IL-2, IL-4, anti-IFNγ and anti-IL-12/23 p40. **B** Schematic overview of AAI model induced by ovalbumin (OVA): cultured Tc1, Tc2, or Th2 OTI/OTII cells were transferred intravenously into WT mice. The next day, mice were challenged i.t. with 50 μg OVA and 10 μg HDM daily for 4 consecutive days. Analysis was performed on day 5. **C** Numbers of total eosinophils, proportions of GR-1% eosinophils, and

numbers of CD4$^+$ or CD8$^+$ T cells determined in BAL by flow cytometry analysis. No cells, OVA + HDM: $n = 4$, OTI Tc1, OVA + HDM: $n = 5$, OTI Tc2, OVA + HDM: $n = 6$, OTII Th2, OVA + HDM: $n = 6$. **D** Quantification of IFNγ$^+$, IL-4$^+$, IL-5$^+$, and IL-13$^+$ Th and Tc cells in BAL by flow cytometry. No cells, OVA + HDM: $n = 5$, OTI Tc1, OVA + HDM: $n = 5$, OTI Tc2, OVA + HDM: $n = 6$, OTII Th2, OVA + HDM: $n = 6$. Symbols represent individual mice, bars indicate mean values ±SEM. *$P < 0.05$, **$P < 0.01$, ***$P < 0.001$, (two-tailed Kruskal–Wallis test corrected for multiple testing). AAI allergic airway inflammation, WT wild type, BAL bronchoalveolar lavage. Source data are provided as a Source Data file.

## Type-2 skewing of Tc cells in vivo is suppressed by IFNγ

To identify the micro-environmental signals underlying Tc2 cell abundance in asthma, we exposed WT mice to inhaled HDM in the presence or absence of recombinant-IFNγ (recIFNγ) (Fig. 4A), which skews the lungs microenvironment towards a mixed type-1/type-2 cytokine milieu. HDM sensitization followed by repetitive HDM challenges induced allergic airway inflammation (AAI), characterized by high numbers of eosinophils, B cells, T cells, and DCs in the BAL (Fig. 4B, Supplementary Fig. 7A). Co-administering recIFNγ during sensitization led to reduced eosinophilia and immune cell infiltration (Fig. 4B, Supplementary Fig. 7A).

While the Th compartment contained more IFNγ$^+$ and IL-17A$^+$ cells in HDM-sensitized mice treated with recIFNγ, type-2 cytokine production capacity of the Th population was unaffected (Fig. 4C, Supplementary Fig. 7B). Tc cells from HDM-sensitized mice showed increased IFNγ levels accompanied by strongly reduced IL-5 and IL-13 production capacity upon treatment with recIFNγ (Fig. 4C, D, Supplementary Fig. 7C). Overall, the Tc cell compartment shifted from a dominant Tc1 phenotype (95.4% Tc1 of all cytokine-producing Tc cells) in PBS-sensitized mice to a highly Tc2-skewed population in HDM-sensitized animals (80.2% Tc2), which was strongly suppressed by recIFNγ (42.1% Tc2; Fig. 4E). A similar, though less prominent shift was found for Th cells (Supplementary Fig. 7D).

To determine whether blocking IFNγ signaling would conversely increase type-2 skewing of Tc cells, we next administered an antibody

to block IFNγ (anti-IFNγ) prior to HDM sensitization (Fig. 4F). Anti-IFNγ did not substantially alter immune cell infiltration in BAL of HDM-sensitized mice, although we observed somewhat reduced numbers of infiltrating T cells (Fig. 4G, Supplementary Fig. 7E). Whereas cytokine production capacity of the Th cell population was not altered by administration of anti-IFNγ, IFNγ levels were decreased and IL-5 production capacity by Tc cells increased in HDM-sensitized mice treated with anti-IFNγ (Fig. 4H, I, Supplementary Fig. 7F). Overall, Tc2-skewing in HDM-treated animals was enhanced by anti-IFNγ (i.e. from 50.6% to 76.1% of all cytokine-producing Tc cells; Fig. 4J). An effect of anti-IFNγ on Th2 skewing was not observed (Supplementary Fig. 7G).

These experiments show that extensive type-2 skewing of lung Tc cells occurs in a mouse model of AAI and that such Tc skewing is suppressed by IFNγ signals within the inflammatory tissue microenvironment.

## Activation of Tc cells by Tnfaip3-deficient cDC1s suppresses type-2 skewing

Type-1 conventional DCs (cDC1) induce Tc cell activation via antigen cross-presentation[35]. DC activation induces NF-κB signaling, which stimulates pro-inflammatory cytokine production. NF-κB activation is negatively regulated via (de)ubiquitination of key signaling proteins by TNFα-induced protein 3 (Tnfaip3 or A20)[36]. Targeted deletion of *Tnfaip3* in cDC1s using a *Tnfaip3*-floxed allele combined with Langerin(Lg)-cre (referred to as *Tnfaip3*$^{Lg-KO}$ mice[37]) was shown to abolish

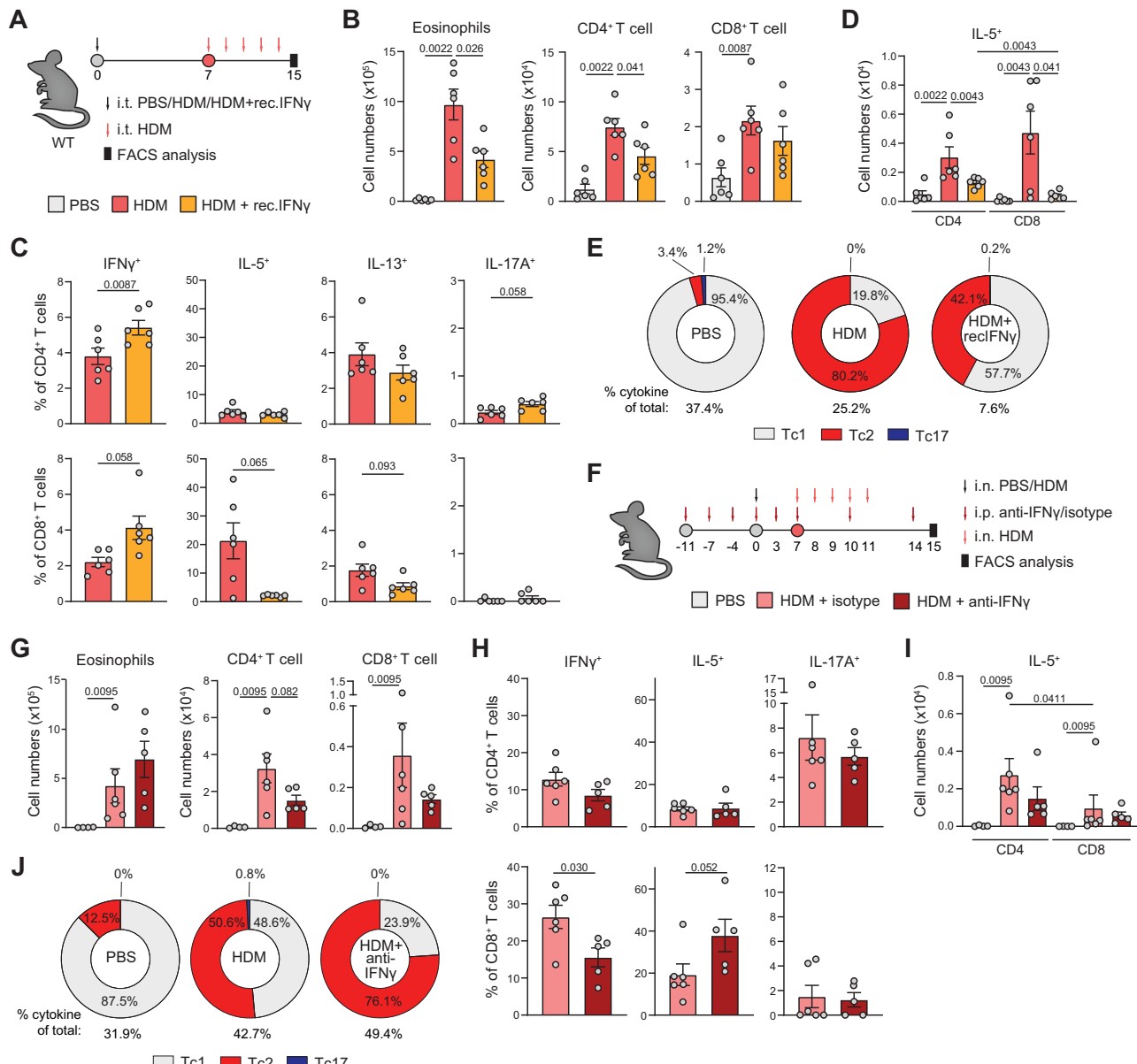

**Fig. 4 | IFNγ suppresses type-2 skewing of Tc cells. A** Schematic overview of AAI model induced by HDM: WT mice were sensitized i.t. with PBS or 10 µg HDM on day 0 and challenged i.t. with 10 µg HDM daily from day 7 to day 11. 50 ng recombinant-IFNγ (recIFNγ) was given i.t. simultaneously with HDM sensitization on day 0. Analysis was performed on day 15. **B** Numbers of eosinophils, CD4+, and CD8+ T cells determined in BAL by flow cytometry analysis. **C** Quantification of IFNγ+, IL-5+, IL-13+, and IL-17A+ Th and Tc cells in BAL by flow cytometry. **D** Numbers of IL-5+ Th and Tc cells in BAL measured by flow cytometry. **E** Pie charts summarizing proportions of Tc1, Tc2, and Tc17 cells in PBS or HDM-sensitized WT mice treated with or without recIFNγ. **F** Schematic overview of AAI model induced by HDM: WT mice were sensitized intranasally (i.n.) with PBS or 1 µg HDM on day 0 and challenged i.n. with 10 µg HDM daily from day 7 to day 11. 0.5 mg anti-IFNγ was given intraperitoneally (i.p.) on days −11, −7, −4, 0, 3, 7, 10, and 14. Analysis was performed

on day 15. **G** Numbers of eosinophils, CD4+, and CD8+ T cells were determined in BAL by flow cytometry. PBS: *n* = 4, HDM: *n* = 6, HDM+anti-IFNγ: *n* = 5. **H** Quantification of IFNγ+, IL-5+, and IL-17A+ Th and Tc cells in BAL by flow cytometry. HDM: *n* = 6, HDM+anti-IFNγ: *n* = 5. **I** Numbers of IL-5+ Th and Tc cells in BAL measured by flow cytometry. PBS: *n* = 4, HDM: *n* = 6, HDM+anti-IFNγ: *n* = 5. **J** Pie charts summarizing proportions of Tc1, Tc2, and Tc17 cells in PBS or HDM-sensitized WT mice treated with or without anti-IFNγ. '% cytokine of total:' indicates percentage of all Tc cells that produce one of the cytokines assayed. Symbols represent individual mice, B–D: *n* = 6 mice per group, G-I: PBS, *n* = 4 mice; HDM, *n* = 6 mice; HDM+anti-IFNγ, *n* = 5 mice. Bars indicate mean values ±SEM. *P < 0.05, **P < 0.01 (two-tailed Mann–Whitney U test). AAI allergic airway inflammation, HDM house dust mite, WT wild type, BAL bronchoalveolar lavage. Source data are provided as a Source Data file.

eosinophilic airway inflammation upon HDM sensitization, whereby lung cDC1s upregulated IL-12 production and Tc cells showed increased IFNγ production[37]. To test the effect of this type-1-skewing Tnfaip3-deficient cDC1 compartment on Tc2 cell induction, we induced HDM-mediated AAI in WT (*Tnfaip3*Lg-WT) and *Tnfaip3*Lg-KO mice (Fig. 5A). Indeed, *Tnfaip3*Lg-KO mice showed severely reduced eosinophilia and immune cell infiltration in their BAL, compared to *Tnfaip3*Lg-WT mice (Fig. 5B, Supplementary Fig. 8A). In line with type-1-skewing,

percentages of IFNγ+ Th and Tc cells were increased in the BAL of HDM-sensitized *Tnfaip3*Lg-KO compared with *Tnfaip3*Lg-WT mice, whereas IL-5 production capacity was significantly reduced only in Tc cells (Fig. 5C). Treatment with anti-IFNγ prior to HDM sensitization in *Tnfaip3*Lg-KO mice restored eosinophilia and immune cell infiltration in BAL (Fig. 5B, Supplementary Fig. 8A)[37]. Importantly, whereas anti-IFNγ in *Tnfaip3*Lg-KO mice restored the percentages of IFNγ+ Th and Tc cells to WT levels, IL-5+ cell proportions were specifically elevated in Tc cells (Fig. 5C, D,

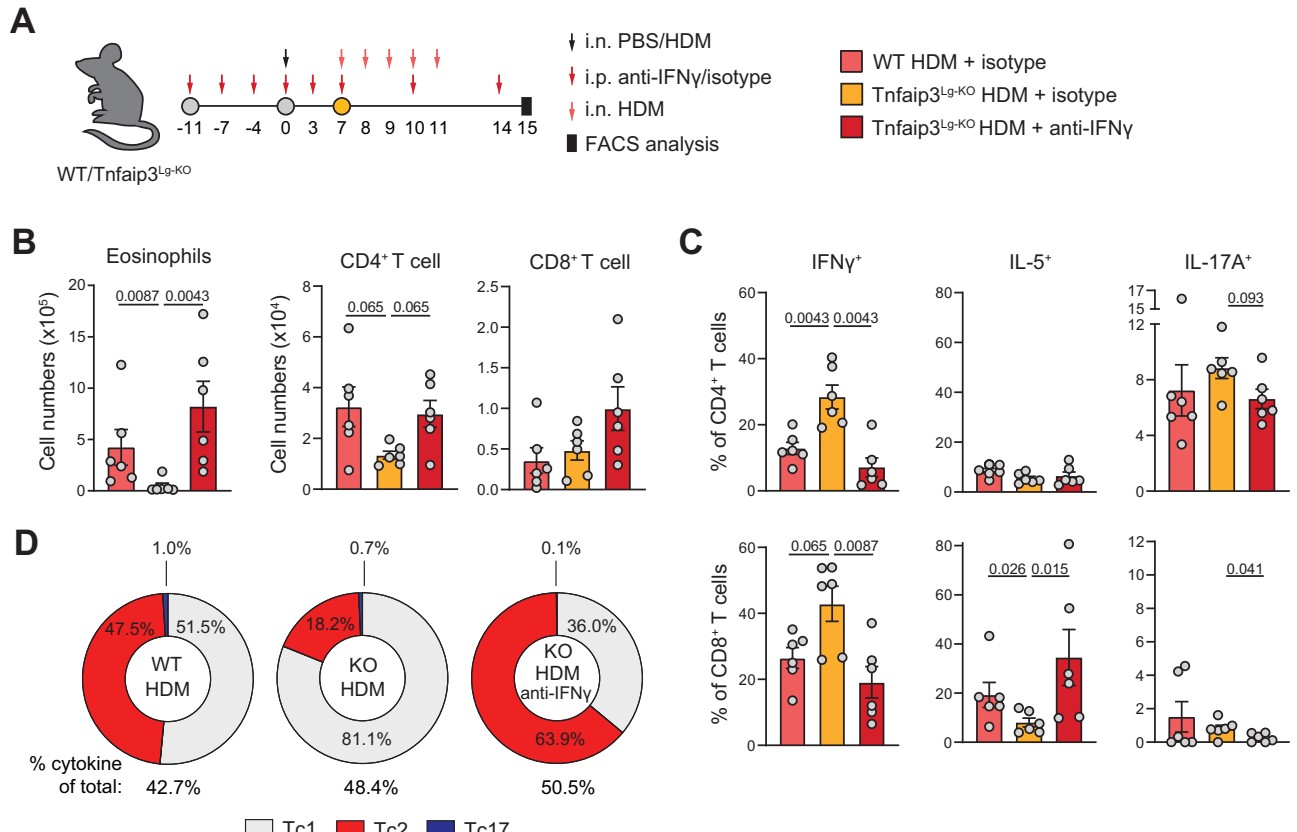

**Fig. 5 | Depletion of *Tnfaip3* in Langerin⁺ cDC1s suppresses type-2 skewing of Tc cells which is restored by blocking IFNγ. A** Schematic overview of AAI model induced by HDM: WT and *Tnfaip3*^Lg-KO mice were sensitized intranasally (i.n.) with PBS or 1 µg HDM on day 0 and challenged i.n. with 10 µg HDM daily from day 7 to day 11. 0.5 mg anti-IFNγ was given intraperitoneal (i.p.) on days −11, −7, −4, 0, 3, 7, 10, and 14. Analysis was performed on day 15. **B** Numbers of eosinophils, CD4⁺, and CD8⁺ T cells determined in BAL by flow cytometry analysis. **C** Quantification of IFNγ⁺, IL-5⁺, and IL-17A⁺ Th and Tc cells in BAL by flow cytometry. **D** Pie charts

summarizing proportions of Tc1, Tc2, and Tc17 cells in PBS or HDM-sensitized WT or *Tnfaip3*^Lg-KO mice treated with or without anti-IFNγ. '% cytokine of total:' indicates percentage of all Tc cells that produce one of the cytokines assayed. Symbols represent individual mice, B-C: *n* = 6 mice per group, bars indicate mean values ±SEM. *P < 0.05, **P < 0.01 (two-tailed Mann–Whitney *U* test). AAI allergic airway inflammation, HDM house dust mite, WT wild type, BAL bronchoalveolar lavage. Source data are provided as a Source Data file.

Supplementary Fig. 8B, C). Of note, IL-17A⁺ Tc cell levels in *Tnfaip3*^Lg-KO mice were reduced upon anti-IFNγ administration (Fig. 5C).

In summary, these findings show that signals from Tnfaip3-deficient cDC1s suppress the formation of Tc2 cells in an IFNγ-dependent manner.

## Type-2 skewing of Tc cells is promoted by the alarmin IL-33

IL-33 is a potent inducer of type-2 immunity that is elevated in the lungs of asthma patients[38]. Since Tc cells require IL-33 for optimal antiviral responses[11], we contemplated a potential role for IL-33 in Tc2 cell development in asthma. To test this hypothesis, we exposed Gata3^YFP/YFPFoxp3^IRES/mRFP reporter mice to IL-33 to induce type-2 airway inflammation (Fig. 6A). Exposure to IL-33 resulted in the accumulation of ILC2 and eosinophils in the BAL, as previously reported (Fig. 6B, Supplementary Fig. 9A)[39]. IL-33 treatment increased both Th and Tc cell numbers, as well as their levels of the IL-33 receptor (IL-33R/ST2) and the canonical type-2 transcription factor GATA3 (Fig. 6B, C). Although IL-33 suppressed IFNγ production by both Th and Tc cells, differential impact on type-2 cytokine production was observed (Fig. 6D, for gating see: Supplementary Fig. 9B). Whereas Th cells only showed increased IL-13 levels upon exposure to IL-33, Tc cells exhibited augmented IL-4 and especially IL-5 production - both in percentages and absolute counts (Fig. 6D, E). Increased proportions of IFNγ⁺/IL-4⁺ and IFNγ⁺/IL-5⁺ Tc cells upon treatment with IL-33, supports elevated type-2 skewing of IFNγ⁺ Tc cells induced by IL-33 (Fig. 6F, Supplementary Fig. 9C). Indeed, both Tc and Th cell compartments shifted

from a dominant type-1 phenotype in PBS-treated mice to a highly type-2-skewed population in IL-33-treated animals (Fig. 6G, Supplementary Fig. 9D).

To investigate the type-2 skewing effect of IL-33 in the context of a primary antigen-specific Tc response, we transferred naïve OTI Tc cells into WT recipient mice and exposed these to OVA + HDM and/or IL-33 (Fig. 7A). In the lung-draining mediastinal lymph node (MLN), OVA + HDM exposure provoked IFNγ but also type-2 cytokine production in OTI Tc cells (Fig. 7B), revealing that naïve Tc cells can be skewed towards Tc2 cells when receiving primary TCR stimulation in lymph nodes. After 5 days, OVA + HDM treatment alone did not induce pulmonary eosinophilia (Fig. 7C). This was in contrast to mice exposed to IL-33 alone or in combination with HDM + OVA, which displayed a significant eosinophilia that likely also involves local ILC2 activation. Importantly, IL-33 significantly enhanced the accumulation of type-2 skewed OTI Tc2 cells in the BAL as compared to IL-33 or OVA + HDM treatment alone (Fig. 7C, D, Supplementary Fig. 10). Alongside potent IL-5 induction, antigen-mediated TCR activation also elicited IL-13 production by Tc2 cells when combined with IL-33 in vivo (Fig. 7D).

To investigate the role of endogenous IL-33 on Tc2 cell formation in an allergen-specific model of AAI, we exposed IL-33-deficient (IL-33^KO) mice to HDM (Fig. 8A). Although lung eosinophilia was established in the absence of IL-33, BAL eosinophil numbers were reduced in HDM-sensitized IL-33^KO mice compared to IL-33^WT animals (Fig. 8B). In contrast, Th, Tc cell and B cell infiltration in the BAL remained

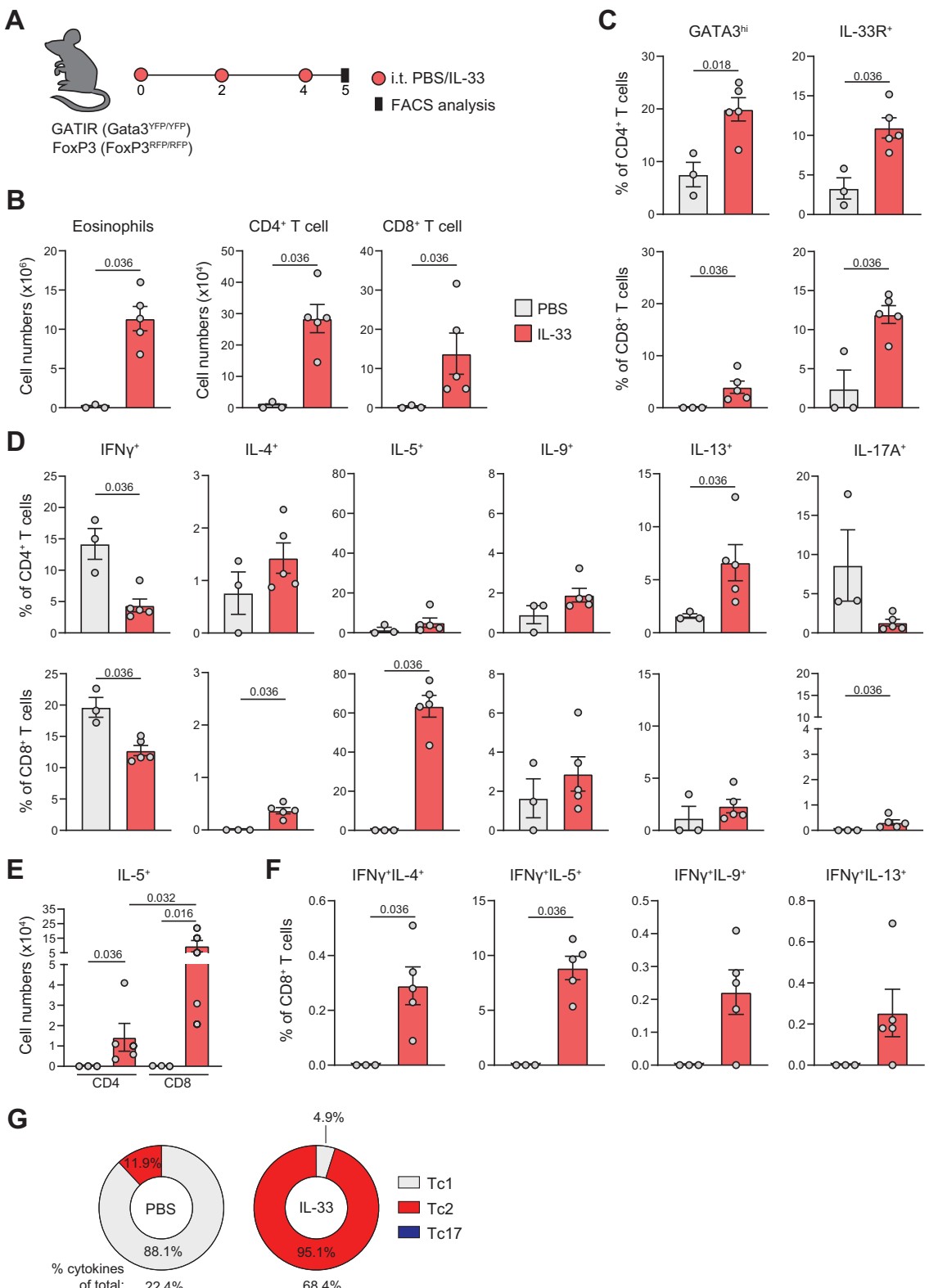

**Fig. 6 | IL-33 induces type-2 skewing of Tc cells. A** Schematic overview of eosinophilic airway inflammation induced by IL-33: GATIR/FoxP3 (*Gata3*[YFP/YFP]*FoxP3*[IRES/mRFP]) mice were treated i.t. with PBS or 0.5 μg IL-33 on day 0, 2 and 4. Analysis was performed on day 5. **B** Numbers of eosinophils, CD4+, and CD8+ T cells determined in BAL by flow cytometry. **C** Quantification of GATA3[hi] and IL-33R+ (ST2) Th and Tc cells in BAL by flow cytometry. **D** Quantification of IFNγ+, IL-4+, IL-5+, IL-9+, IL-13+, and IL-17A+ Th and Tc cells in BAL by flow cytometry. **E** Numbers of IL-5+ Th and Tc cells in the BAL measured by flow cytometry. **F** Quantification of IFNγ+IL-4+, IFNγ+IL-5+, IFNγ+IL-9+, and

IFNγ+IL-13+ double-producing Tc cells in BAL. **G** Pie charts summarizing proportions of Tc1, Tc2, and Tc17 cells in PBS or IL-33 treated WT mice. '% cytokine of total:' indicates percentage of all Tc cells that produce one of the cytokines assayed. Symbols represent individual mice, PBS: *n* = 3 mice, IL-33: *n* = 5 mice, bars indicate mean values ±SEM. *$P$ < 0.05, (two-tailed Mann–Whitney U test). AAI allergic airway inflammation, WT wild type, BAL bronchoalveolar lavage. Source data are provided as a Source Data file.

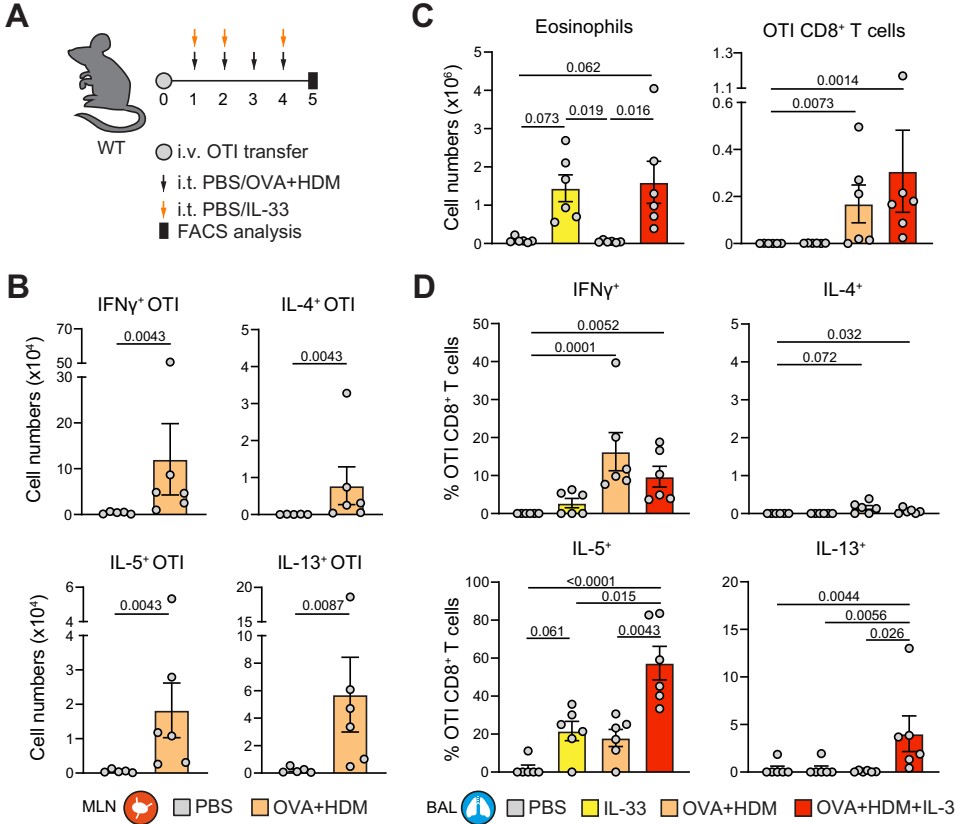

**Fig. 7 | Type-2 skewing by IL-33 in the context of an antigen-specific Tc response. A** Schematic overview of eosinophilic airway inflammation induced by IL-33 and/or ovalbumin (OVA): naïve OTI cells were first transferred into WT recipient mice. The next day, mice were challenged intratracheally with 50 µg OVA and 10 µg HDM, 0.5 µg IL-33, or a combination treatment. Analysis was performed on day 5. **B** Numbers of IFNγ[+], IL-4[+], IL-5[+], and IL-13[+] OTI Tc cells were determined in the mediastinal lymph node (MLN) by flow cytometry. PBS: $n = 5$, OVA + HDM: $n = 6$

mice. **C** Numbers of eosinophils and OTI Tc cells determined in BAL by flow cytometry. **D** Quantification of IFNγ[+], IL-4[+], IL-5[+], and IL-13[+] OTI Tc cells in BAL by flow cytometry. Symbols represent individual mice, **C**, **D** $n = 6$ mice per group, bars indicate mean values ±SEM. *$P < 0.05$, **$P < 0.01$, ***$P < 0.001$ (two-tailed Mann−Whitney or Kruskal−Wallis test corrected for multiple testing). WT wild type, BAL bronchoalveolar lavage. Source data are provided as a Source Data file.

unaltered (Fig. 8B, Supplementary Fig. 11A). Whereas frequencies of IFNγ, IL-4, IL-13, and IL-17A producing Th or Tc cells were not affected by the absence of IL-33, IL-9[+] cells were reduced in both of the populations (Fig. 8C, Supplementary Fig. 11B, C). Notably, Tc cells showed a striking loss of IL-5 positivity in both percentage and absolute cell counts, which was not seen in Th cells (Fig. 8C, D). Moreover, we observed a substantial loss of IFNγ[+] Tc cells concomitantly producing IL-5/IL-9 in IL-33[KO] mice (Supplementary Fig. 11D). Hence, IL-33 was critical to establish a dominant type-2 skewed lung Tc population in HDM-driven AAI, whereas the impact of IL-33 on the Th cell phenotype was only modest (Fig. 8E, Supplementary Fig. 11E).

Together, these data show that IL-33 promotes the formation of a dominant type-2 skewed lung Tc population in various models of eosinophilic airway inflammation – irrespective of the presence or absence of an antigen-specific T-cell activation signal.

### IL-33 promotes Tc2 formation in an indirect manner, likely via broad induction of IL-4

To investigate whether IL-33 directly induces Tc2 cell formation we exposed naïve Tc cells isolated from WT mice to combinations of IL-4 – a known inducer of IL-5 and IL-13 production by Tc cells[13,40]−and IL-33 in the presence of TCR stimulation. GATA3, IL-5, and IL-13 levels were increased in Tc cells by the addition of IL-4, but not by IL-33 exposure alone (Fig. 9A, B). Interestingly, under these conditions co-treatment with IL-4 and IL-33 did not further enhance the type-2 skewing capacity of IL-4 alone, despite robust induction of IL-33R (ST2) expression by IL-4 (Fig. 9A, B).

These results suggest that IL-33 might act in an indirect manner on Tc2 cell formation in vivo, possibly by increasing IL-4 levels within the tissue microenvironment. Since IL-4 can be produced by various immune cells, we measured intracellular IL-4 levels in a wide range of immune cell types isolated from the lungs of mice exposed to IL-33 (see Fig. 6A for treatment protocol). We observed a substantial increase in IL-4[+] Th cells, B cells, DCs, basophils, and particularly eosinophils induced by IL-33 treatment (Fig. 9C). Hence, IL-33 evokes a broad induction of IL-4 synthesis by various immune cells with key roles in eosinophilic airway inflammation, which in turn is likely to fuel type-2 skewing of Tc cells.

## Discussion

Here we provide compelling evidence that the Tc cell compartment in asthma undergoes substantial phenotypic skewing, resulting in a functional shift away from type-1 (e.g., IFNγ) and towards type-2 (e.g., IL-5, IL-9) cytokine production. The increased presence of circulating Tc2 cells in asthma patients associates with more severe disease and corticosteroid insensitivity, and is particularly evident during an acute disease exacerbation. Mechanistically, we use mouse models of AAI to show that Tc2 cell generation is strongly dependent on IL-33 in vivo, whereas IFNγ and cDC1 activity suppress this process.

Given that exacerbations are predominantly triggered by respiratory viral infections[41], Tc cells have attracted surprisingly little attention as mediators of exacerbations. Although previous reports have also linked Tc cells to type-2 cytokine production in (severe) asthma[18,21,40], we now show that during disease

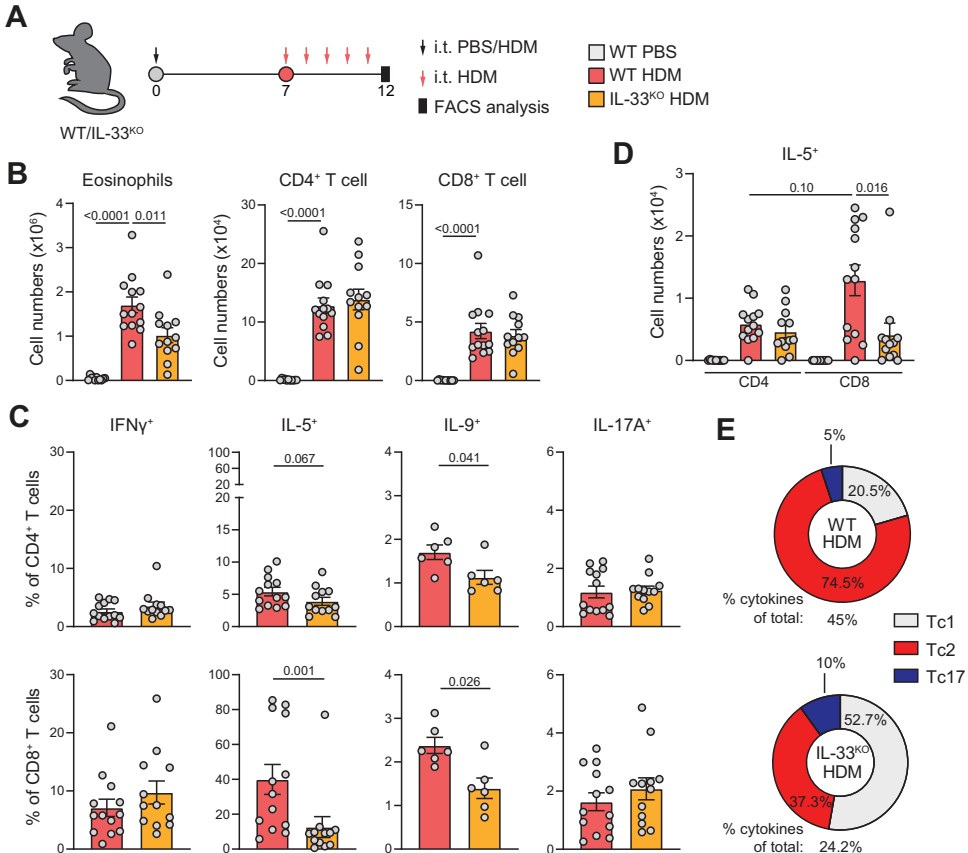

**Fig. 8 | Loss of IL-33 reduces type-2 skewing in Tc cells. A** Schematic overview of AAI model induced by HDM: WT and IL-33^{KO} mice were sensitized i.t. with PBS or 10 µg HDM on day 0 and challenged i.t. with 10 µg HDM daily from day 7 to day 11. Analysis was performed on day 12. **B** Numbers of eosinophils, CD4+, and CD8+ T cells determined in BAL by flow cytometry analysis. PBS: $n = 11$, WT HDM: $n = 13$, IL-33^{ko} HDM: $n = 12$ mice. **C** Quantification of IFNγ+, IL-5+, IL-9+, and IL-17A+ Th and Tc cells in BAL by flow cytometry. For IFNγ, IL-5 and IL-17A, WT HDM: $n = 13$ and for IL-33^{ko} HDM: $n = 12$ mice. For IL-9, $n = 6$ mice per group. **D** Numbers of IL-5+ Th and Tc cells in the BAL measured by flow cytometry. PBS: $n = 11$, WT HDM: $n = 13$, IL-33^{ko} HDM: $n = 12$ mice. **E** Pie charts summarizing proportions of Tc1, Tc2, and Tc17 cells in PBS or HDM-sensitized WT and IL-33^{ko} mice. '% cytokine of total:' indicates percentage of all Tc cells that produce one of the cytokines assayed. Symbols represent individual mice, bars indicate mean values ±SEM. *$P < 0.05$, ***$P < 0.001$, ****$P < 0.0001$ (two-tailed Mann–Whitney $U$ test). AAI allergic airway inflammation, HDM house dust mite, WT wild type, BAL bronchoalveolar lavage. Source data are provided as a Source Data file.

exacerbations type-2 cytokine production capacity by circulating Tc cells is approximately doubled to ~25% of the total population. This is in line with increased levels of IL-13+ Tc cells in BAL of mice on an experimental asthma exacerbation protocol[42], as well as higher numbers of IL-4+ Tc cells in lung tissue from asthma patients who died from an exacerbation[17]. Furthermore, we also show that type-2 cytokine production capacity of Tc cells can rival or sometimes even surpasses the capacity of Th2 cells in (exacerbating) asthma patient samples and in mouse models of AAI.

What about the origin of the expanding type-2 cytokine-producing Tc cells in severe asthma? Although the majority of Tc cells produce IFNγ, we consistently detected Tc2 cells in healthy controls, confirming that non-canonical cytokine production capacity is a native feature of the steady-state Tc compartment[43]. Increased abundance of Tc2 cells in severe asthma could therefore result from the expansion of existing cells or through phenotypic skewing of the more abundant canonical Tc1 cells. Our data support a significant contribution of the latter scenario, as we detected increased levels of Tc cells producing IFNγ together with a type-2 cytokine in asthma patients with severe disease or in mice with AAI, likely representing an intermediate state of Tc1 cells adopting a type-2 effector phenotype. The concept of Tc plasticity in type-2 inflammatory disease receives additional support from observations in mouse studies[13,14], which implicated epigenetic priming of the Th2 cytokine loci by GATA3 as a requisite for Tc1-to-Tc2

conversion[14]. Whether human Tc plasticity in asthma patients also has an epigenetic basis remains to be determined.

Whether Tc responses in asthma are antigen-specific remains unclear. Tc cells recognizing the Der-p1 HDM antigen have been linked to severe atopic disease[44], in line with an allergen-specific Tc response via cross-presentation of HDM antigens by DCs. Using the OVA-OTI experimental system, we indeed show that allergen-specific priming of naive Tc cells in the lymph node can lead to the formation of Tc2 cells (Fig. 10). Alternatively, cytokines can activate (virus-specific) Tc cells in an antigen-independent manner[13]. This is supported by our experiments showing that IL-33 exposure induces Tc2 cell expansion within 5 days in the absence of specific antigens, although concomitant TCR stimulation could further augment Tc2 cell formation. In mice, IL-4 has been identified as a key driver of IL-5 and IL-13 expression in Tc cells[13,40], although IL-4 alone only induced weak type-2 cytokine expression in cultured human Tc cells[45]. The alarmin IL-33 is now recognized as a critical initiator of type-2 immunity[46] and genetic polymorphisms near *IL33* or *IL1RL1* (encoding the IL-33-R) are strongly associated with asthma susceptibility[47]. Importantly, IL-33 also drives antiviral Tc cell responses[11], is released by human lung epithelial cells upon infection with rhinovirus[48,49], is elevated in BAL samples from asthma patients[38], and blocking IL-33 signals ameliorates asthma exacerbation symptoms in mouse models[49,50]. We demonstrate that IL-33 is a key inducer of type-2 cytokine production in Tc cells in vivo,

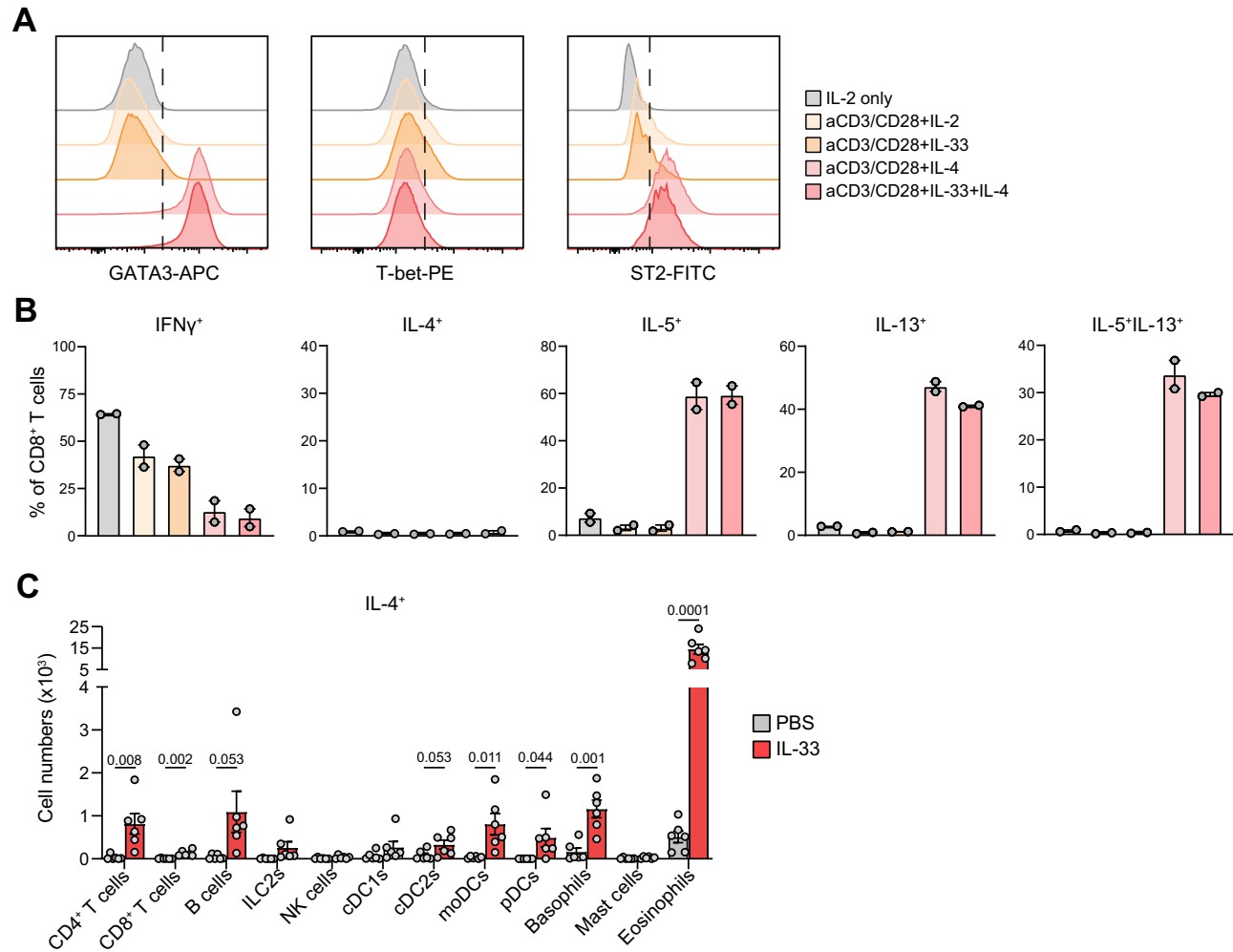

**Fig. 9 | Evidence for indirect type-2 skewing of Tc cells by IL-33. A** Flow cytometric characterization of the transcription factors GATA3 and T-bet and the IL-33 receptor ST2 on cultured mouse Tc cells exposed to the indicated cytokines in combination with TCR stimulation via anti-CD3/CD28 beads. **B** Quantification of IFNγ+, IL-4+, IL-5+, IL-9+, IL-13+, and IL-5+IL-13+ cultured Tc cells measured by flow cytometry (n = 2). **C** Numbers of IL-4 producing CD4+ and CD8+ T cells, B cells, group 2 innate lymphoid cells (ILC2s), natural killer (NK) cells, conventional DC (cDC) 1 and cDC2 cells, monocyte-derived DCs (moDCs), plasmacytoid DCs (pDCs), basophils, mast cells and eosinophils determined in BAL of PBS or IL-33 treated mice (n = 6 mice per group, treatment regime shown in Fig. 6A) by flow cytometry analysis. Symbols represent individual mice, bars indicate mean values ±SEM. *P < 0.05, **P < 0.01, ***P < 0.001 (two-tailed Mann–Whitney U test). BAL bronchoalveolar lavage. Source data are provided as a Source Data file.

providing a direct link between respiratory viral infections, Tc1-to-Tc2/Tc9 plasticity and asthma exacerbations (Fig. 10). We envision that the asthmatic lung and draining lymph node tissue microenvironment promotes the differentiation of Tc2 cells through chronic IL-4 signaling. In the lung, this is further augmented by epithelial release of IL-33 upon viral infection to further boost Tc1 plasticity and trigger an episode of increased symptom severity. In such a model, our current evidence suggests that IL-33 promotes Tc2 formation in an indirect manner, presumably via a potent induction of IL-4 secretion in the lungs by Th2 cells but also, for example, eosinophils (Fig. 10). In lung-draining lymph nodes IL-33 also induces Tc2 skewing, which could be mediated via DC or ILC2 activation and may involve IL-4 production by for example follicular Th cells[51,52]. Additional experiments using cell type-specific deletion of IL-4 and/or IL-33R/ST2 in vivo are required to further dissect these mechanisms.

IL-33 thus appears to play a central role in severe asthma by promoting the activation of not only Th2 cells, but also inflammatory CD45RO+ ILC2s[7] and Tc2 cells. The latter two cell types may be particularly relevant for poor responses to corticosteroids, as both CD45RO+ ILC2s and Tc cells were reported to be less sensitive to suppression by steroids than Th cells[7,53,54] (Fig. 10). Our data

reveal that Tc2 cell levels are increased in severe asthma patients, including those requiring high doses of steroids to combat symptoms. These findings suggest that a subset of severe, therapy-resistant asthma patients may be characterized by elevated Tc2 cell formation and susceptible to treatment with biologics targeting IL-4 or IL-33 or components in their downstream signaling pathways[2]. Future studies focused at the in-depth molecular characterization of Tc2 cells and their role in asthma will likely provide valuable new insight into the pathophysiology of uncontrolled disease. Such studies could have broader implications, since Tc2 cell activity has been implicated in various other diseases, including psoriasis, fibrosis, and cancer[43,55,56].

Previous studies have shown that murine cDC1s can suppress type-2 inflammation upon infection with helminths[57] or exposure to HDM[58]. Our findings upon *Tnfaip3* deletion in Langerin+ cDC1s corroborate these results and demonstrate that cDC1 activity suppresses IL-5 production by Tc, but not Th cells, resulting in reduced IL-5-mediated eosinophilic inflammation. Additionally, we show that increased IFNγ levels only decrease IL-5 synthesis in Tc cells, whereas neutralizing IFNγ specifically promoted IL-5 production in Tc cells. Conversely, the absence of IL-33 in the lungs of HDM-exposed mice only restrains

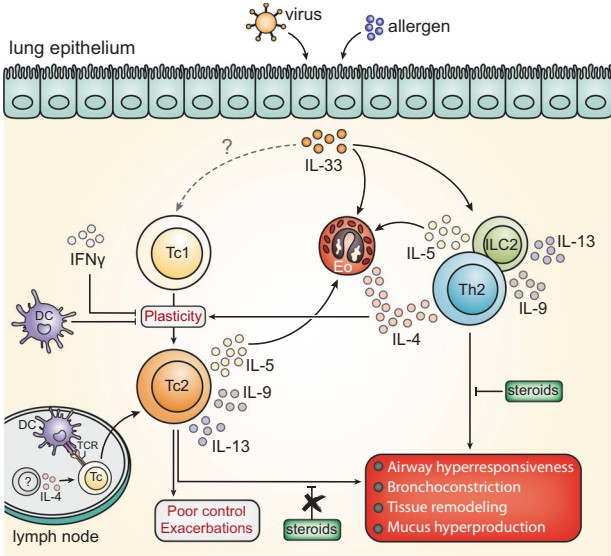

**Fig. 10 | Tc2 cell formation as a key mediator of uncontrolled asthma and exacerbations.** Schematic drawing summarizing the main findings of this study and placing them in the context of asthma pathophysiology (adapted from Ref. 66.). Allergens induce epithelial release of alarmins such as IL-33, which can promote type-2 cytokine production (IL-4, IL-5, IL-9, and IL-13) by allergen-specific CD4+ Th2 cells that are classically considered to fuel the hallmark symptoms of allergic asthma (listed in the red box). In addition, IL-33 promotes local IL-4 production by various other immune cells, including eosinophils. Respiratory viruses, the major triggers of asthma exacerbations, promote CD8+ Tc cell activation in the lung and also induce epithelial IL-33 release. In severe asthmatics, in particular during a virus-induced exacerbation, high levels of IL-33 promote immune cell-derived IL-4 production to induce phenotypic skewing of the Tc1 cell population towards type-2 cytokine-producing Tc2 cells. It remains possible that IL-33 also directly acts up Tc1 cells to activate type-2 cytokine synthesis in vivo. Type-2 skewing likely occurs via plasticity of canonical IFNγ-producing Tc1 cells and is suppressed by IFNγ and cDC1 activity. Tc2 formation can also occur in draining lymph nodes in the context of a primary antigen-specific Tc response, although the local cellular source of IL-4 in this scenario remains to be determined. Altogether, this Tc2-derived surge in type-2 cytokine production capacity may explain the increased disease burden, exacerbation frequencies, and corticosteroid resistance observed in severe asthma patients.

type-2 skewing in the Tc cell compartment. Together, these findings reveal that - compared to Th cells - the phenotype of Tc cells is substantially more malleable in response to changes in the microenvironment. While this increased sensitivity to an altered cytokine milieu may allow Tc cells to functionally adapt and safeguard homeostasis[59], it could also render them more susceptible to adopting a pathogenic phenotype in the context of chronic inflammation.

In conclusion, we show that the formation of Tc2 cells in AAI is regulated by the alarmin IL-33, most likely via the induction of plasticity in canonical type-1 Tc cells. IFNγ signaling and cDC1 activity are able to suppress type-2 cytokine production by Tc cells. Type-2 skewing of Tc cells is associated with uncontrolled asthma and reaches peak levels during disease exacerbations, supporting an important role for the aberrant functional adaptation of Tc cells in the pathogenesis of severe type-2 respiratory disease.

## Methods

### Study design
The goal of this study was to investigate the role of type-2 cytokine-producing CD8+ T (Tc2) cells in severe asthma. We performed flow cytometry analyses on peripheral blood mononuclear cell (PBMC) fractions from healthy controls and various types of asthma patients to identify and characterize Tc2 cells. To generate new insight into the

mechanisms that promote Tc2 generation in vivo, we induced AAI in mice by sensitizing them to house dust mite. A combination of treatments (recIFNγ, anti-FNγ or IL-33) and transgenic animals (IL-33$^{KO}$, OTI/OTII, $Tnfaip3^{Lg-KO}$) were used to alter the lung microenvironment or Tc cell activation, which was followed by flow cytometry analyses to characterize the Tc cell compartment.

### Patients and healthy controls
Asthma patients and healthy controls were recruited by the Franciscus Gasthuis & Vlietland hospital in Rotterdam as previously described[7]. Cohort characteristics are summarized in Table S1. Smokers (>10 pack years), obese patients (BMI > 35) and patients whom received systemic corticosteroid therapy three months prior to inclusion were excluded. Sex/gender (self-reported) was not considered for the recruitment of patients. Healthy controls were age- and sex-matched. Based on their asthma control questionnaire (ACQ) score[31], asthma patients were subdivided into the following groups: controlled (ACQ < 0.75), partially controlled (ACQ = 0.75–1.5), and uncontrolled (ACQ > 1.5) asthma. Secondly, patients were also subdivided based on the registered number of exacerbations in the year prior to inclusion. Lastly, patients were subdivided into low (0-500 dose equivalent units), medium (500–1000 bioequivalent units) and high (>1000 bioequivalent units) inhaled steroid intake groups by converting their individual steroid dosages to bioequivalent units according to recently published guidelines[60]. Atopic asthma was defined by the presence of a clinically relevant allergy. T2 asthma was defined according to GINA 2020 guidelines: the presence of blood eosinophils ≥150/μl and/or forced expiratory nitrogen oxide (FeNO) ≥20 ppb and/or the presence of a clinically relevant allergy. All patients and healthy controls provided a written informed consent (including publication of indirect identifiers) and all experimental procedures were reviewed and approved by the Medical Ethical Committees of the Franciscus Gasthuis & Vlietland and the Erasmus MC. Buffy coats from healthy volunteers were provided by the Sanquin Blood Bank (informed consent obtained by Sanquin).

### Mice
All mice were housed and bred under SPF conditions at the Erasmus MC and analyzed at 6-12 weeks of age after being euthanized by an overdose of pentobarbital. Housing temperature was between 19-24 °C, humidity was between 40-70% and mice were placed on a fixed light (7AM–19PM)–dark cycle. In all experiments, we strived for a 1:1 male/female ratio; non-transgenic littermates were used as WT controls. Experimental and control animals were co-housed. WT and OTI/OTII (carrying an OVA-specific transgenic TCR) mice were purchased from Envigo (WT strain: #680) and Jackson (OTI strain: #003831, OTII strain: #004194) respectively. $Tnfaip3^{fl/fl}$ mice were crossed to $Cd207^{CRE/+}$ (Langerin-CRE) mice to generate $Tnfaip3^{Lg-KO}$ mice[37]. The $Gata3^{YFP/YFP}$ (GATIR) mouse strain[61] was crossed to the $Foxp3$-IRES-mRFP (FIR) reporter mice strain[62] to obtain $Gata3^{YFP/YFP}Foxp3^{IRES/mRFP}$ mice. IL-33$^{KO}$ mice[63] were crossed with $Gata3^{YFP/YFP}Foxp3^{IRES/mRFP}$ mice. All mice were backcrossed to the C57BL/6 genetic background for at least six generations. Please contact the corresponding author if interested in obtaining these mouse strains. All experiments were performed with approval by the animal ethics committee of the Erasmus MC under a permit licensed by the Dutch government (AVD101002016637).

### Induction of eosinophilic airway inflammation in vivo
For adoptive transfer studies of Tc1/Tc2/Th2 cells, naive CD4+ and CD8+ T cells were isolated from spleen and lymph nodes of OTI/OTII mice using a CD4 or CD8 T-cell MACS isolation kit (Miltenyi Biotec) and subsequently sorted using a FACSAria (BD Biosciences), Cells were selected on negativity for DAPI (Invitrogen)

and further gated as CD8$^+$CD62L$^+$CD44$^-$. Sorted cells were cultured in 96-well round-bottom plates precoated with 5 µg/mL anti-CD3 (BD Biosciences, 145-2C11) and 3 µg/mL anti-CD28 (BD Biosciences, 37.51) in T-cell medium (IMDM containing 10% FCS, $5 \times 10^{-5}$ M β-mercaptoethanol, 1× GlutaMAX, and 55 µg/mL gentamicin; Lonza) for 7 days in the presence of IL-2 (5 ng/mL, R&D) for Tc1 polarization, or IL-2, IL-4 (50 ng/mL; PeproTech) and anti-IFNγ (10 µg/mL; BD Biosciences B27) for Tc2 polarization. For Th2 polarization, cells were cultured in the presence of IL-2, IL-4 (10 ng/mL), anti-IFNγ (5 µg/mL) and anti-IL-12/23 p40 (5ug/mL, BD Biosciences C17.8). Cells were splitted at day 3 and day 5. The cytokines were added again when cells were splitted. A total of $3 \times 10^6$ cultured T cells were transferred intravenously (i.v.) into WT recipients. The next day, mice were exposed intratracheally (i.t.) for 4 consecutive days to PBS or 50 µg OVA (Endotoxin-free, Endograde) and 10 µg HDM (Greer). Animals were euthanized 1 day later for FACS analyses. WT mice and $Tnfaip3^{\text{Lg-KO}}$ mice were sensitized intranasally (i.n.) with 1 µg HDM (Greer) or 40 µl PBS (GIBCO Life Technologies) as a control on day 0[37]. Next, mice were challenged i.n. on days 7-11 with 10 µg HDM and euthanized on day 15. $Gata3^{\text{YFP/YFP}}Foxp3^{\text{RFP/RFP}}$ and WT mice were treated three times i.t. every other day with 0.5 µg IL-33 (BioLegend, USA) or PBS, as described previously[64]. These mice were euthanized on day 5. For adoptive transfer of naive OTI cells, naive CD8$^+$ T cells were isolated from spleen and lymph nodes of OTI mice using a CD8 T-cell MACS isolation kit (Miltenyi Biotec). A total of $3 \times 10^6$ cultured T cells were transferred i.v. into WT recipients. The next day, mice were exposed intratracheally (i.t.) for 4 consecutive days to PBS or 50 µg OVA (Endotoxin-free, Endograde) and 10 µg HDM (Greer), and 3 times with 0.1 or 0.5 µg IL-33 (BioLegend, USA). Animals were euthanized 1 day later for FACS analyses. IL-33$^{\text{KO}}$ and WT mice were sensitized i.t. with 10 µg HDM or 80 µl PBS as a control on day 0, and challenged i.t. on days 7-11 with 10 µg HDM, followed by sacrificing the animals on day 12. During HDM, IL-33, and OVA exposures, all mice were anesthetized using isoflurane.

### Antibody treatment of mice during HDM-induced AAI

Recombinant-IFNγ (50 ng, R&D systems) was given i.t. to WT mice simultaneously with HDM sensitization. $Tnfaip3^{\text{Lg-KO}}$ and WT mice were treated with anti-IFNγ (0.5 mg per i.p. injection, clone XMG1.2) on days −11, −7, −4, 0, 3, 7, 10, and 14. An isotype control antibody was used for experiments with anti-IFNγ (GL113, 0.5 mg per i.p. injection).

### Isolation of T cells

For all mice in vivo experiments, BAL were obtained by flushing the lungs three times with 1 mL PBS containing 0.5 mM EDTA (Sigma-Aldrich). Human PBMCs were isolated from asthma patient blood draws or buffy coats as previously described[65]. Red blood cells were lysed using osmotic lysis buffer (8.3% NH$_4$Cl, 1% KHCO$_3$, and 0.04% NA$_2$EDTA in Milli-Q water). To purify human CD8 T cells for cultures, PBMCs from buffy coats were first depleted for CD4$^+$, CD14$^+$, CD16$^+$, CD19$^+$, CD36$^+$, and CD235ab$^+$ cells by magnetic cell sorting (MACS) using MojoSortTM streptavidin Nanobeads (BioLegend, USA) and LS Columns (Miltenyi Biotec). After MACS, cells were rested overnight in RPMI-1640 medium supplemented with 5% fetal calf serum (FCS) at 4 °C. The next morning, cells were extracellularly stained with antibodies for 30 mins at 4 °C and with LIVE/DEADTM Fixable aqua for 15 mins at 4 °C. CD8 T cells were sorted by fluorescence-activated cell sorting (FACS) using a FACSAriaTM III (BD).

### Culture of T cells

Sorted human CRTH2$^-$ and CRTH2$^+$ CD8 T cells were cultured in RPMI-1640 containing GlutaMAX-I (Thermo Fisher Scientific) supplemented with 10% FCS. 40.000 cells per well were cultured in 96-round-bottom plates and stimulated for 4 days with soluble anti-CD3/anti-CD28 (10 µg/mL, BioLegend, USA), IL-2 (10 U/mL), IL-7 (20 ng/mL), and anti-IFNγ (5 µg/mL, BioLegend, USA). For mouse cultures, naive CD8$^+$ T cells were isolated from spleen and lymph nodes of WT mice by depletion of CD11b$^+$, CD11c$^+$, GR1$^+$, Ter119$^+$, NK1.1$^+$, and CD19$^+$ cells using MACS. The cell suspension was then subsequently sorted on a FACSAriaTM III (BD) by selecting negatively for DAPI (Invitrogen) and gating on CD8$^+$CD62L$^+$CD44$^-$. Sorted cells were cultured in 96-well round-bottom plates precoated with 5 µg/mL anti-CD3 (BD Biosciences, 145-2C11) and 3 µg/mL anti-CD28 (BD Biosciences, 37.51) in T-cell medium (IMDM containing 10% FCS, $5 \times 10^{-5}$ M β-mercaptoethanol, 1× GlutaMAX, and 55 µg/mL gentamicin; Lonza) for 7 days in the presence of IL-2 (5 ng/mL, R&D), IL-4 (50 ng/mL; PeproTech), IL-33 (50 ng/mL; BioLegend, USA). and anti-IFNγ (10 µg/mL; BD Biosciences B27). Cells were splitted at day 3 and the cytokines were added again. Cells were analyzed by flow cytometry as described below.

### RNA isolation and quantitative PCR

RNA was extracted from ~500,000 naive (direct after sort) or in vitro polarized CD4+ and CD8 + OTII/OTI T cells using the RNeasy Micro kit (Qiagen) according to manufacturer's instructions. For quantitative PCR, RNA was synthesized into cDNA using RevertAid H Minus Reverse Transcriptase and random hexamer primers in the presence of RiboLock RNAse inhibitor (Thermo Fisher Scientific). Quantitative PCR was performed using 7.5 µL 2x SYBR Green Universal Master Mix (Life Technologies) and 6 pmol of both forward and reverse primer, added to a total reaction volume of 15 uL using nuclease-free water. Transcript levels were normalized to those of the $Cript$ housekeeping gene. Primer sequences used in this study can be found in Table S3.

### ELISA

IL-5 and IL-13 levels were measured in culture supernatants by enzyme-linked immunosorbent assay (ELISA, Thermo Fisher Scientific) according to the manufacturer's protocols.

### Flow cytometry

Table S4 lists antibodies to human proteins used for phenotyping of PBMCs. Monoclonal antibodies used for mouse flow cytometry analyses are listed in Table S5. For both human and mouse: dead cells were excluded using Fixable viability dye (eBioscience). Cells were stained extracellular for 30 mins at 4 °C. For intracellular cytokine stainings, cells were stimulated with PMA (10 ng/mL; Sigma) and Ionomycin (500 nM; Merck) in the presence of Golgi-Stop (BD) for 4 hours at 37 °C. Cells were fixed with PFA (2%, 10 mins, 4 °C) and permeabilized with Saponin (0.5%, 30 mins at RT). For mouse intracellular transcription factor measurements, cells were fixed and permeabilized using the eBioscience Foxp3/transcription factor staining buffer set (Thermo Fisher Scientific) according to the manufacturer's instructions. For analysis by flow cytometry, data were acquired on an LSR II flow cytometer (BD) or an LSRSymphony (BD) using FACSDiva v9 (BD) software and analyzed with FlowJo (Tree Star Inc., USA) software.

### Dimensionality reduction (tSNE) analysis

Using the tSNE FlowJo v10 plugin, we performed an Unbiased hierarchal t-Distributed Stochastic Neighbor Embedding (tSNE) analysis. Individual patient fcs files were imported into FlowJo v10. CD4$^+$ and CD8$^+$ T cells were pre-gated and of these Th and Tc cells, all events that were negative for any cytokine were gated out. The 'cytokine-positive' CD4 and CD8 gates of all asthma patients were concatenated using compensated parameters to form two files: one containing all cytokine-positive CD8$^+$ T cells and one containing all cytokine-positive CD4$^+$ T cells. These files were then

down-sampled to 45000 total events. tSNE was run for 300 iterations with the following selected parameters: CD45RA, CRTH2, IL-4, IL-5, IL-9, IL-13, IL-17A, IFNγ.

### Statistical analyses

Statistical significance was determined using a two-tailed Mann–Whitney U test, Pearson correlation coefficient analysis, two-tailed Wilcoxon rank-sum test, or two-tailed Kruskal-Wallis test with GraphPad Prism software (v8). A $p$ value of <0.05 was considered statistically significant.

### Reporting summary

Further information on research design is available in the Nature Portfolio Reporting Summary linked to this article.

## Data availability

Source data depicted in all figures are provided with this paper (see Source data file.xlsx). Raw flow cytometry, quantitative PCR, and ELISA data files generated and analyzed for the current study are available from the corresponding author upon reasonable request. Source data are provided with this paper.

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

## Acknowledgements

We thank all members from the Stadhouders and Hendriks laboratories for helpful discussions. R.W.H. is supported by Dutch Lung Foundation grant 4.1.18.226. R.S. is supported by a Dutch Research Council Vidi grant (09150172010068), an Erasmus MC Fellowship, and a Dutch Lung Foundation Junior Investigator grant (4.2.19.041JO). D.H. was supported by extra funds coordinated with the Erasmus MC Executive Board and joint BIG funding (between Erasmus University Rotterdam and Erasmus MC). H.V. is supported by a Dutch Lung Foundation grant (9.2.15.065FE).

## Author contributions

E.K.vd.P.: design of the work; acquisition, analysis, and interpretation of data; writing the manuscript. L.K.: design of the work; acquisition and analysis of data. H.V.: design of the work; acquisition and analysis of data. M.vN.: acquisition of data. M.J.W.dB.: acquisition of data. G.M.dB.: acquisition of data. I.M.B.: acquisition of data. Mirjam Kool: design of the work. G.A.T.: design of the work. G.J.B.: design of the work. DH: design of the work. RWH: design of the work; interpretation of data; writing the manuscript, project supervision. R.S.: design of the work; interpretation of data; writing the manuscript, project supervision.

## Competing interests

G.J.B. and G.A.T. declare the following competing interests: G.J.B. has received grant/research support for consultations and/or speaking at conferences from Novartis, GSK, AstraZeneca, ALK, Teva, Sanofi, and Chiesi. G.A.T. has received research support and consultation fees from OM Pharma, AstraZeneca, and Chiesi, all paid to a research foundation. The remaining authors declare no competing interests.
