## [Peer Review File · Nature Communications]

Type-2 CD8⁺ T cell formation relies on interleukin-33 and is linked to asthma exacerbationsREVIEWER COMMENTS

Reviewer #1 (Remarks to the Author):

General comments:

In this manuscript, the authors examined the prevalence of CD8+ T cells that produce type 2 cytokines (Tc2 cells) in peripheral blood of patients with asthma as well as bronchoalveolar lavage (BAL) specimens in mouse models of airway inflammation. Tc2 cells and their roles in type 2 airway inflammation have been described previously by other investigators. However, our knowledge is still limited regarding Tc2 cells as compared to conventional Th2-type CD4+ T cells. Therefore, subject matter of this manuscript is potentially interesting and important. The strength of the manuscript is a comprehensive analysis between clinical phenotypes of asthma and cytokine production by Tc2 cells. However, several major concerns are identified regarding the biological significance.

Specific comments:

1. The main tool used in this project is intracellular cytokines staining and FACS analysis of T cells after stimulation with PMA plus ionomycin. As PMA plus ionomycin is a powerful stimulus, they may represent the capacity to T cells to produce cytokines, but may not reflect physiologic responses. It will be useful to have some experiments to determine how these capacities to produce cytokines in Tc2 cells can be translated to actual production of cytokines in more physiologic settings.
2. In Figure 3 (line 130), the authors conclude that quantitative increase in Tc2 cell frequencies rival those seen for Th2 cells during asthma exacerbation. How can the authors compare the amounts of type 2 cytokines produced by Tc2 cells and Th2 cells?
3. In Figure 6, the authors administered mice with IL-33 and analyzed Th2 cells and Tc2 cells in BAL fluids. It would be useful to have representative FACS plots of intracellular cytokine staining for these mouse experiments. In addition, it is rather strange that T cells were analyzed in a condition where IL-33 was administered without antigens. The experiments need to be repeated with antigens to optimize TCR-dependent activation and proliferation of Th2 cells and Tc2 cells.
4. In Figure 7, the authors analyzed the frequency of Th2 cells and Tc2 cell in mice that had been exposed to HDM. IL-5-producing Tc2 cells are clearly detected in BAL fluids (Fig. 7C). However, a major question remains regarding their biological functions. Are these Tc2 cells play roles in airway inflammation and immunopathology of the lungs? Can you delete CD8+ T cells or use CD8-deficient mice to investigate the immunological roles for Tc2 cells?

Reviewer #2 (Remarks to the Author):

This manuscript studies the production of type 2 cytokines by CD8+ T cells (Tc2), and provides correlative evidence that CD8+ T cells can participate to the pathogenesis of asthma in humans. Mouse experiments provide evidence that DC1 and IL-33 participate to the induction of Tc2, and the results are consistent with Tc2 aggravating the disease. The work is convincing and the experiments are well controlled.

The lack of direct evidence that Tc2 (or at least CD8+ T cells) are involved in the pathology is lacking. Does depletion of CD8+ T cells in mice affect the induction or severity of lung inflammation in the dust mite model? (depletion of CD8+ T cells could be obtained genetically or using specific antibodies in vivo). Or can the pathology be induced or aggravated by after adoptive transfer of Tc2 cells? Direct evidence that Tc2 are involved in the pathology would strengthen the manuscript.

Reviewer #3 (Remarks to the Author):

This manuscript describes the presence of type-2 CD8+ T cells (Tc-2) in asthma patients and links an increase in the prevalence of these cells with asthma severity and steroid resistance. Under allergic conditions, the authors attributed IL-33 as a critical factor for Tc-2 cell generation. Also, the authors proposed that IFN-g as a negative regulator of Tc-2 generation and deletion of Tnfrsf3 in cDC1 cells suppress the Tc-2 conversion.

Strength:

The main strength of the manuscript is identifying IL-33 as a critical inducing factor for Tc-2 cells. There are some reports on IL-33 as an innate factor for type 2 cytokine induction in CD4 T cells without TCR signaling (Guo et al., 2015; 26322482). This study may strengthen the hypothesis that IL-33 may serve as a critical factor for switching Tc cells into Tc-2 cells.

Weakness:

Some of the experiments are already reported. The current study mainly focuses on Tc-2 cell prevalence in asthma patients, whereas IL-33-mediated Tc-2 cell conversion is the novel part of the story.

Comments:

The initial part of the study investigates the prevalence of Tc-2 cells in asthma patients. Correlating Tc-2 cells with asthma severity and skewing of Tc1 cells into Tc-2 cells is interesting, but several groups have already reported this observation (references below), and these results do not add novelty to the study. IL-4 is known to be a skewing factor for the conversion of Tc cells to Tc-2 cells. Therefore, the increased Tc-2 number in uncontrolled patients could be due to an indirect effect of more Th2 and ILC2 cells.

The results related to the mouse model of HDM-induced pulmonary inflammation and IL-5+, IL-13+ Th cell generation are not convincing. Wild-type mice challenged with HDM should show an increase in the percentage of IL-5+ and IL-13+ Th cells. However, experiments in this study show the same level of IL-5+ Th cells and a decrease in the percentage of IL-13+ Th cells (Fig. 4C) with HDM treatment. Also, the data on role of IL-33 in Tc-2 cell induction are too preliminary to make a conclusion. The increase in Tc-2 cells in response to IL-33 challenge could be due to an indirect effect caused by severe pulmonary inflammation mediated by type 2 immune responses. Additional experiments are needed to determine if IL-33 is a critical factor for Tc-2 cell generation. This is essential given that the experiments related to IL-33 are the main strength and novelty in this study.

Major concerns:

1. In the first three figures, the authors report that Tc-2 cell numbers increase in asthma patients and correlate with asthma severity and steroid resistance. However, several reports have shown an increase in type 2 cytokine-producing CD8+ T lymphocytes associates with asthma severity (Hilvering et al., 2018; 29907870, Gelfand et al., 2017; 28658551, Gelfang and Hinks, 2019; 31376407). These results do serve as an objective for hypothesis and supportive information to the current study, but summarizing them in a single figure rather than three figures is sufficient.
2. There are several limitations with HDM-induced airway inflammation results. The HDM challenges should increase the percentage of IL-5+ and IL-13+ Th cells. However, Fig. 4C shows that HDM did not increase the percentage of IL-5+ Th cells, and IL-13+ Th cells were decreased in percentage. These results do not establish Tc-2s as a major producer of type-2 cytokines under allergic conditions. More data are needed, or the conclusions need to be changed.
3. IFN-g is known to be a negative regulating factor for the Th2 response in the mouse model of allergy, and studies show the negative effect of IFN-g on Tc-2 cells (e.g., Apte et al., 2010; 20562261). The current form of Fig. 4 does not add any new information. The authors should examine if IFN-g negatively effects IL-33 release or IL-33 induced Tc-2 cell skewing in vitro.
4. In Figs. 6 and 7, the authors proposed that IL-33 induces Tc-2 cell skewing, and IL-33 deficiency reduces the Tc-2 skewing of Tc cells in response to HDM. Although IL-33 deficiency reduces Tc-2 cells, it is unclear whether this is due to a direct effect on CD8 T cells, or an indirect

effect of IL-4 secreted from Th2 cells and ILC2s in response to IL-33 treatment. The transition of IFN-g producing CD8+ T cells into IL-5 and IL-13 producing Tc-2 by IL-4 treatment has been reported (Jia et al., 2013; 23509358). An in vitro experiment of Tc-2 cells skewing with IL-33 is required to confirm this hypothesis. Because IL-33 and IL-4 might have cooperative effects with IL-4 initiated and IL-33 amplified or vice versa, an in vitro experiment with various combinations of IL-4 and IL-33 to show the conversion of Tc into Tc-2 is needed to see if this supports the hypothesis.

5. The decreased Tc-2 cells in Tnfaip3Lg-KO mice might be due to the lower availability of type 2 cytokines induced by Th cells, as in Fig. S7B. Lower availability of IL-4 may lead to the less conversion of Tc cells into Tc-2 cells. Nevertheless, if the Tnfaip3Lg-KO mice have low IL-33 levels in the BALF, this would at least be consistent with the author's hypothesis.

6. The study lacks mechanistic experiments to determine how IL-33 induces Tc-2 cell generation. The conclusion is drawn based on the preliminary observation of decreased Tc-2 cell numbers in IL-33 deficient mice. RNA-seq of IL-33 treated CD8+ T cells to identify factors responsible for Tc-1 to Tc-2 conversion and other mechanistic studies seem important.

7. The presence of up to 60% IFN-g+ Th cells at a steady state in Fig. 4C is surprising. Mouse models have mainly naïve Th cells and do not produce IFN γ at steady state.

8. In Fig. 4C, there is a large difference in IFN-g+ Th cells between the PBS challenged group (20-60%) vs mice challenged with HDM (1.5-3%). However, Fig. S6B shows similar cell numbers between these two groups. This must be resolved.

9. There is a huge variation in the percentage of IFN γ + Th cells and Tc cells with the PBS group in Figs. 4C and 6D.

Overall, the paper is very descriptive, and some things were known. The interesting data relate to IL-33, but even here conclusions cannot be drawn with certainty. The authors need more definitive data to strengthen their conclusions.

NCOMMS-22-01394; Van der Ploeg et al. - Author Response

General statement:

We thank the three reviewers for the interest in our study and their constructive feedback. Addressing their comments has strengthened our main conclusions and improved the quality of the study. We have added a substantial amount of new data to the revised manuscript, including 3 new main figures and 2 new supplementary figures. Key additions to the revised manuscript include:

1. New evidence supporting the capacity of Tc2 cells to autonomously induce eosinophilic airway inflammation *in vivo*.
2. Data revealing an interplay between TCR stimulation and IL-33 in promoting Tc2 formation *in vivo*.
3. *In vitro* and *in vivo* experiments that further explore how IL-33 promotes Tc2 cell formation mechanistically.

Please find our point-by-point responses below. Those parts of the manuscript that are new or were modified have been highlighted in yellow.

Reviewer: 1

General comments:

In this manuscript, the authors examined the prevalence of CD8⁺ T cells that produce type 2 cytokines (Tc2 cells) in peripheral blood of patients with asthma as well as bronchoalveolar lavage (BAL) specimens in mouse models of airway inflammation. Tc2 cells and their roles in type 2 airway inflammation have been described previously by other investigators. However, our knowledge is still limited regarding Tc2 cells as compared to conventional Th2-type CD4⁺ T cells. Therefore, subject matter of this manuscript is potentially interesting and important. The strength of the manuscript is a comprehensive analysis between clinical phenotypes of asthma and cytokine production by Tc2 cells. However, several major concerns are identified regarding the biological significance.

Response: We thank the reviewer for his/her positive overall assessment of our study.

Specific comments:

1. The main tool used in this project is intracellular cytokines staining and FACS analysis of T cells after stimulation with PMA plus ionomycin. As PMA plus ionomycin is a powerful stimulus, they may represent the capacity to T cells to produce cytokines, but may not reflect physiologic responses. It will be useful to have some experiments to determine how these capacities to produce cytokines in Tc2 cells can be translated to actual production of cytokines in more physiologic settings.

Response: This is an important point indeed. We therefore conducted several experiments with both mouse and human Tc cells in which we measure cytokine mRNA expression and protein production directly after TCR stimulation, omitting any subsequent PMA/ionomycin stimulation. In these experiments, human Tc cells exhibited identical qualitative and quantitative cytokine expression profiles as seen after PMA/ionomycin restimulation (**Reviewer Fig.1A**). Furthermore, gene expression profiling and ELISA showed that mRNA and secreted protein levels of type-2 cytokines are increased after TCR stimulation without PMA/ionomycin in mouse and human Tc cells (**Reviewer Fig.1B-D**). We added a paragraph to the manuscript (**page 3, line 61-68**) to discuss these findings, which have been integrated into **Fig.3** and **Supplementary Fig.2**.

Reviewer Fig.1. (A) Quantification of IFN γ , IL-5, IL-9 and IL-13 production by CRTh2⁻ and CRTh2⁺ Tc cells. Cells were isolated from human peripheral blood and cultured for 4 days with IL-2, anti-CD3 and anti-CD28. Hereafter, cells were incubated with golgi-stop with or without PMA/ionomycin (denoted by - and +, respectively) for 4 hours and cytokine production was measured using intracellular flow cytometry. **(B)** Quantitative PCR analysis of *Ifng*, *Il4*, *Il5*, *Il9*, and *Il13* mRNA levels in mouse splenic Tc cells stimulated *in vitro* for 6 days with IL-2 and anti-CD3/CD28 supplemented with or without IL-4. **(C)** mRNA levels of *IFNG*, *IL4*, *IL5*, *IL9*, and *IL13* in human effector memory Tc cells before and after 24 hours of stimulation with IL-2 and anti-CD3/CD28, as quantified by RNA-Sequencing. Data presented in panel C were obtained from Calderon et al.¹ **(D)** IL-5 and IL-13 levels measured by ELISA in supernatant from naïve OTII CD4⁺ and OTI CD8⁺ T cells cultured in 96-well round-bottom plates precoated with 5 μ g/mL anti-CD3 and 3 μ g/mL anti-CD28 for 7 days in the presence of IL-2 for Tc1 polarization, or IL-2, IL-4 and anti-IFN γ for Tc2 polarization. For Th2 polarization, cells were cultured in the presence of IL-2, IL-4, anti-IFN γ and anti-IL-12.

2. In Figure 3 (line 130), the authors conclude that quantitative increase in Tc2 cell frequencies rival those seen for Th2 cells during asthma exacerbation. How can the authors compare the amounts of type 2 cytokines produced by Tc2 cells and Th2 cells?

Response: We based this conclusion on a comparison of percentages of cytokine-positive cells (i.e. 'frequencies'), but agree with the reviewer that this does not necessarily reflect similar amounts of cytokine production on a per cell basis. To expand on this, we used our flow cytometry data to calculate geometric mean fluorescence intensity (gMFI) values for Tc2 and Th2 cells from healthy controls, stable asthma patients

and patients during an active exacerbation (**Reviewer Fig.2**). This analysis revealed that on a per cell basis, Tc cells on average produce less IL-4 and IL-5, but more IL-13 than Th cells. IL-9 levels were similar between both T cell compartments, and all trends were similar for both stable asthma and exacerbation samples. Together, we conclude that while relative frequencies of Tc2 cells in asthma rival those seen for Th2 cells, the amounts of cytokines produced on a per cell basis can differ between Tc2 and Th2 cells - depending on the specific cytokine. We discuss these findings in the revised manuscript (**page 5, line 126-127**), and the data are now included in **Supplementary Fig.5F**.

Reviewer Fig.2. gMFI of IL-4⁺, IL-5⁺, IL-9⁺ and IL-13⁺ Th and Tc cells in peripheral blood samples from healthy controls (HC), stable asthma patients (Asthma) and asthma patients during an active exacerbation (Exa). *P<0.05, **P<0.01, ***P<0.001, ****P<0.0001 (Wilcoxon-paired test).

3. In Figure 6, the authors administered mice with IL-33 and analyzed Th2 cells and Tc2 cells in BAL fluids. It would be useful to have representative FACS plots of intracellular cytokine staining for these mouse experiments.

Response: We have now included the requested flow cytometry gating strategies to identify Th2 and Tc2 cells in all our *in vivo* models for eosinophilic airway inflammation (examples shown in **Reviewer Fig.3**). These plots have been included in the revised manuscript (**Supplementary Fig.6E/7B/9B/11B**).

Reviewer Fig.3. Flow cytometry gating strategies used to identify Th2 and Tc2 cells in BAL fluid. **(A)** Gating strategy for OVA+HDM treated mice. **(B)** Gating strategy for HDM sensitized mice treated with either recIFN γ or anti-IFN γ . **(C)** Gating strategy for IL-33 treated mice. **(D)** Gating strategy for HDM sensitized IL-33^{KO} mice.

In addition, it is rather strange that T cells were analyzed in a condition where IL-33 was administered without antigens. The experiments need to be repeated with antigens to optimize TCR-dependent activation and proliferation of Th2 cells and Tc2 cells.

Response: We agree with the reviewer that intuitively one might not expect T cell activation without antigenic exposure. However, several studies have documented an ‘innate’ capacity (i.e. TCR/antigen-independent) of T cells to be activated by cytokines (e.g. PMID: 26322482, 22665806^{2, 3}). Nevertheless, we do find the reviewer’s proposal relevant for our understanding of Tc2 formation *in vivo*. We therefore repeated our original IL-33 experiment but this time transferred OTI transgenic naive Tc cells carrying an ovalbumin/OVA-specific TCR. Treating these animals with only OVA+HDM, IL-33 alone or IL-33 together with OVA+HDM allows us to investigate Tc2 skewing in primary lymph node antigen-specific (i.e. against OVA) Tc responses. Moreover, we could compare a situation of innate Tc activation (IL-33 only) with an antigen-specific response in the presence of IL-33 (**Reviewer Fig.4**). This revealed that allergen-specific TCR stimulation can induce type-2 cytokine production by naive Tc cells in lung-draining lymph nodes. IL-33 induced eosinophilia and when combined with OVA/HDM significantly enhanced the accumulation of type-2 skewed IL-5⁺/IL-13⁺ OTI Tc2 cells in the BAL as compared to IL-33 or OVA/HDM treatment alone. Alongside potent IL-5 induction, antigen-mediated TCR activation also elicited IL-13 production by Tc2 cells when combined with IL-33 *in vivo*.

These results show that IL-33 promotes Tc2 formation in various models of eosinophilic airway inflammation - irrespective of the presence or absence of an antigen-specific T cell activation signal. However, antigen-mediated TCR stimulation can further augment type-2 skewing of Tc cells by IL-33. These results and conclusions have been included in the revised manuscript (page 7-8, line 222-232/244-246; page 9, line 294-299; page 10, line 310-315; Fig.7 and Supplementary Fig.10).

Reviewer Fig.4. (A) Schematic overview of eosinophilic airway inflammation induced by IL-33 and/or ovalbumin (OVA): naïve OTI cells were first transferred into wildtype (WT) recipient mice. The next day, mice were challenged intratracheally with 50 μ g OVA and 10 μ g HDM, 0.5 μ g IL-33 or a combination treatment. Analysis was performed on day 5. **(B)** Numbers of IFN γ ⁺, IL-4⁺, IL-5⁺, and IL-13⁺ OTI Tc cells determined in the mediastinal lymph node (MLN) by flow cytometry. **(C)** Numbers of eosinophils and OTI Tc cells determined in the bronchoalveolar lavage (BAL) by flow cytometry. **(D)** Quantification of IFN γ ⁺, IL-4⁺, IL-5⁺, and IL-13⁺ OTI Tc cells in BAL by flow cytometry. Symbols represent individual mice, n=4-6 mice per group, bars indicate mean values \pm SEM. *P<0.05, **P<0.01, ***P<0.001 (Mann-Whitney or Kruskal-Wallis test corrected for multiple testing).

4. In Figure 7, the authors analyzed the frequency of Th2 cells and Tc2 cell in mice that had been exposed to HDM. IL-5-producing Tc2 cells are clearly detected in BAL fluids (Fig. 7C). However, a major question remains regarding their biological functions. Are these Tc2 cells play roles in airway inflammation and immunopathology of the lungs? Can you delete CD8⁺ T cells or use CD8-deficient mice to investigate the immunological roles for Tc2 cells?

Response: The reviewer makes an important point here. We were hesitant to deplete CD8⁺ T cells using anti-CD8 antibodies as this might result in incomplete (i.e. tissue-resident memory Tc cells) or unwanted (e.g. CD8⁺ dendritic cells, CD8⁺ gamma-delta T cells) immune cell depletion. Similarly, making use of CD8-deficient mice might be confounded by possible complex compensatory mechanisms. To provide conclusive evidence that Tc2 cells can autonomously trigger eosinophilic airway inflammation *in vivo*, we chose to instead conduct adoptive transfer experiments with antigen-specific Tc2 (and Tc1 as well as Th2) cells. These cells were obtained from *in vitro* skewing of naïve OTI Tc cells or OTII Th cells (**Reviewer Fig.5A-E**), allowing us to specifically activate Tc2/Th2 cells *in vivo* using OVA after their transfer into naïve wildtype recipient animals. These experiments revealed that Tc2 - but not Tc1 - cell transfer and activation *in vivo* induced a marked infiltration of GR-1⁺ inflammatory eosinophils in the airways - highly similar to the inflammation evoked by Th2 cells (**Reviewer Fig.5F-G**). We believe these data provide strong support for a bona fide *in vivo* capacity of Tc2 cells to drive eosinophilic airway inflammation. These experiments have been included in the revised manuscript (page 5-6, line 140-154; Fig.3 and Supplementary Fig.6).

Reviewer Fig.5. (A) Schematic overview of *in vitro* culture protocol of splenic OTI/OTII T cells to generate Tc1, Tc2 and Th2 cells. Naïve OTI or OTII cells were cultured with anti-CD3, anti-CD28 in the presence of IL-2 for Tc1 polarization, or IL-2, IL-4 and anti-IFN γ for Tc2 polarization. For Th2 polarization, cells were cultured in the presence of IL-2, IL-4, anti-IFN γ and anti-IL-12. **(B)** Schematic overview of AAI model induced by ovalbumin (OVA): Tc1, Tc2 or Th2 OTI/OTII cells were transferred i.v. into WT mice. The next day mice were challenged i.t. with 50 μ g OVA and 10 μ g HDM daily for 4 days. Analysis was performed on day 5. **(C)** Flow cytometry analysis of intracellular IFN γ and IL-13 levels in cultured OTI/OTII cells. **(D)** Quantitative PCR analysis of *Gata3*, *Tbet*, *Ifng*, *Il4*, *Il5*, and *Il13* present in cultured OTI/OTII cells. **(E)** IL-5 and IL-13 levels measured by ELISA in supernatant from cultured OTII Th and OTI Tc cells. **(F)** Numbers of total eosinophils, proportions of GR-1% eosinophils, and numbers of CD4⁺ or CD8⁺ T cells determined in BAL by flow cytometry analysis. **(G)** Quantification of IFN γ ⁺, IL-4⁺, IL-5⁺, and IL-13⁺ OTI Th and OTII Tc cells in BAL by flow cytometry. Symbols represent individual mice, n=5-6 per group, bars indicate mean values \pm SEM. *P<0.05, **P<0.01, ***P<0.001, (Mann-Whitney U test). AAI, allergic airway inflammation; WT, wild type; BAL, bronchoalveolar lavage.

Reviewer: 2

This manuscript studies the production of type 2 cytokines by CD8⁺ T cells (Tc2), and provides correlative evidence that CD8⁺ T cells can participate to the pathogenesis of asthma in humans. Mouse experiments provide evidence that DC1 and IL-33 participate to the induction of Tc2, and the results are consistent with Tc2 aggravating the disease. The work is convincing and the experiments are well controlled.

Response: We are glad to read that the reviewer believes our data are convincing and generated through well-controlled experimental approaches.

The lack of direct evidence that Tc2 (or at least CD8⁺ T cells) are involved in the pathology is lacking. Does depletion of CD8⁺ T cells in mice affect the induction or severity of lung inflammation in the dust mite model? (depletion of CD8⁺ T cells could be obtained genetically or using specific antibodies in vivo). Or can the pathology be induced or aggravated by after adoptive transfer of Tc2 cells? Direct evidence that Tc2 are involved in the pathology would strengthen the manuscript.

Response: We fully agree with this comment. As was stated in our response to Reviewer 1 (Comment #4, see above), we have now included adoptive transfer experiments with OVA-specific Tc2 cells that provide conclusive evidence that activated Tc2 - but not Tc1 - cells can autonomously trigger a profound eosinophilic airway inflammation (**Reviewer Fig.5**). Importantly, the capacity of Tc2 cells to induce eosinophilia was highly similar to that of Th2 cells, demonstrating that type-2 cytokine production by Tc2 cells is sufficient to drive eosinophilic airway inflammation *in vivo*. These experiments have been included in the revised manuscript (**page 5-6, line 140-154; Fig.3 and Supplementary Fig.6**).

Reviewer: 3

This manuscript describes the presence of type-2 CD8⁺ T cells (Tc-2) in asthma patients and links an increase in the prevalence of these cells with asthma severity and steroid resistance. Under allergic conditions, the authors attributed IL-33 as a critical factor for Tc-2 cell generation. Also, the authors proposed that IFN-g as a negative regulator of Tc-2 generation and deletion of Tnfrsf10b in cDC1 cells suppress the Tc-2 conversion.

Strength:

The main strength of the manuscript is identifying IL-33 as a critical inducing factor for Tc-2 cells. There are some reports on IL-33 as an innate factor for type 2 cytokine induction in CD4 T cells without TCR signaling (Guo et al., 2015; 26322482). This study may strengthen the hypothesis that IL-33 may serve as a critical factor for switching Tc cells into Tc-2 cells.

Weakness:

Some of the experiments are already reported. The current study mainly focuses on Tc-2 cell prevalence in asthma patients, whereas IL-33-mediated Tc-2 cell conversion is the novel part of the story.

Comments:

The initial part of the study investigates the prevalence of Tc-2 cells in asthma patients. Correlating Tc-2 cells with asthma severity and skewing of Tc1 cells into Tc-2 cells is interesting, but several groups have already reported this observation (references below), and these results do not add novelty to the study. IL-4 is known to be a skewing factor for the conversion of Tc cells to Tc-2 cells. Therefore, the increased Tc-2 number in uncontrolled patients could be due to an indirect effect of more Th2 and ILC2 cells.

The results related to the mouse model of HDM-induced pulmonary inflammation and IL-5⁺, IL-13⁺ Th cell generation are not convincing. Wild-type mice challenged with HDM should show an increase in the percentage of IL-5⁺ and IL-13⁺ Th cells. However, experiments in this study show the same level of IL-5⁺ Th cells and a decrease in the percentage of IL-13⁺ Th cells (Fig. 4C) with HDM treatment. Also, the data on role of IL-33 in Tc-2 cell induction are too preliminary to make a conclusion. The increase in Tc-2 cells in response to IL-33 challenge could be due to an indirect effect caused by severe pulmonary inflammation mediated by type 2 immune responses. Additional experiments are needed to determine if IL-33 is a critical factor for Tc-2 cell generation. This is essential given that the experiments related to IL-33 are the main strength and novelty in this study.

Response: We thank the reviewer for his/her thorough assessment of our study and agree that identifying IL-33 as an important player in Tc2 formation is a key strength of our work. Below we address in detail the concerns mentioned here.

Major concerns:

1. In the first three figures, the authors report that Tc-2 cell numbers increase in asthma patients and correlate with asthma severity and steroid resistance. However, several reports have shown an increase in type 2 cytokine-producing CD8⁺ T lymphocytes associates with asthma severity (Hilvering et al., 2018; 29907870,

Gelfand et al., 2017; 28658551, Gelfang and Hinks, 2019; 31376407). These results do serve as an objective for hypothesis and supportive information to the current study, but summarizing them in a single figure rather than three figures is sufficient.

Response: We agree with the reviewer that Tc2 cells have previously been linked to asthma disease severity, as also referred to in our Introduction. However, we do believe that the data we showed in the original **Fig.1-3** offers more than the previously reported increased Tc(2) levels in asthmatics and the correlation with symptom severity. We believe our analyses of the human Tc/Th cell compartments in asthma contain two important novel aspects: 1) a comprehensive type-2 cytokine characterization of Tc/Th cells in asthma, including key cytokine IL-9 (see recent findings in for example PMID: 32532832⁴) that shows the strongest and most consistent correlations with disease severity as well as therapy resistance, and 2) the first analysis of Tc/Th cell cytokine production in the context of asthma exacerbations, including analysis of samples obtained during exacerbations. To accommodate the reviewer's request, we attempted to merge the data describing specifically these two novel aspects in a single figure, but ended up with a very large and unwieldy figure. We therefore chose to divide the data over two smaller distinct figures (new **Fig.1-2**). However, if the reviewer still prefers a single main figure for all human data we are of course willing to instead use a single large figure.

2. There are several limitations with HDM-induced airway inflammation results. The HDM challenges should increase the percentage of IL-5⁺ and IL-13⁺ Th cells. However, Fig. 4C shows that HDM did not increase the percentage of IL-5⁺ Th cells, and IL-13⁺ Th cells were decreased in percentage. These results do not establish Tc-2s as a major producer of type-2 cytokines under allergic conditions. More data are needed, or the conclusions need to be changed.

Response: Type-2 responses and eosinophilia in HDM-induced type-2 airway inflammation strictly depend on the generation of a T cell-driven adaptive immune response (see for example Fig.3 from PMID: 27062360, in which we abrogated peripheral T cell activation in a HDM model⁵). In our experience with *in vivo* HDM models (e.g. PMID: 22539286, 27062360, 27939673, 28111308, 29111218, 32255764^{5, 6, 7, 8, 9, 10}) one has to interpret the percentages of cytokine producing T cells while taking into account changes in absolute T cell counts. In HDM models, one expects to see a strong increase in the absolute number of lung-infiltrating IL-5⁺ or IL-13⁺ Th cells, which is exactly what we consistently observe in our current study (see **Supplementary Fig.7C/F, Supplementary Fig.8B, Fig.8D**). To make this clearer, we now included absolute numbers of IL-5 producing Tc and Th cells in the revised version of **Fig.4** (see **panels D & I**). For establishing allergic airway inflammation, it is of secondary importance whether the percentage of cytokine-positive Th cells in the population also increases, since the absolute increase in type-2 cytokine producing T cells will induce eosinophilia. Importantly, our data revealed similar increases in absolute numbers of type-2 cytokine producing Th and Tc cells (see **Fig.4D/I, Supplementary Fig.7C/F, Supplementary Fig.8B, Fig.8D**), strongly suggesting that both Th2 and Tc2 cells are major producers of type-2 cytokines and substantially contribute to eosinophilic airway in our HDM/IL-33/OVA models. In the case of Tc2 cells, we do see significant increases in percentages of IL-5⁺ cells upon HDM/IL-33/OVA exposure, which together with reduced IFN γ ⁺ cells indicates population-level Tc1-to-Tc2 skewing.

3. IFN-g is known to be a negative regulating factor for the Th2 response in the mouse model of allergy, and studies show the negative effect of IFN-g on Tc-2 cells (e.g., Apte et al., 2010; 20562261). The current form of Fig. 4 does not add any new information. The authors should examine if IFN-g negatively effects IL-33 release or IL-33 induced Tc-2 cell skewing *in vitro*.

Response: While we agree that the role of IFN γ as a suppressor of Th2 responses is well described, *in vivo* experiments demonstrating that IFN γ also affects Tc2 cell levels in the context of allergic airway inflammation were still lacking - a knowledge gap that we address in **Fig.4**. The Apte et al. paper referred to indeed points in that direction, but this study was performed in an anti-tumor immune response model and not in an allergen-driven *in vivo* model of Tc2 skewing. Regarding the request to test the effect of IFN γ on *in vitro* Tc2 skewing mediated by IL-33, we refer to our response below (comment #4). Our newly performed experiments point towards an indirect effect of IL-33 on Tc2 formation via a broad induction of IL-4 *in vivo*, with IL-33 by itself not showing any capacity to induce Tc2 formation *in vitro*. Hence, we could not examine the effects of IFN γ on IL-33-induced type-2 skewing *in vitro*.

4. In Figs. 6 and 7, the authors proposed that IL-33 induces Tc-2 cell skewing, and IL-33 deficiency reduces the Tc-2 skewing of Tc cells in response to HDM. Although IL-33 deficiency reduces Tc-2 cells, it is unclear

whether this is due to a direct effect on CD8 T cells, or an indirect effect of IL-4 secreted from Th2 cells and ILC2s in response to IL-33 treatment. The transition of IFN-g producing CD8⁺ T cells into IL-5 and IL-13 producing Tc-2 by IL-4 treatment has been reported (Jia et al., 2013; 23509358). An *in vitro* experiment of Tc-2 cells skewing with IL-33 is required to confirm this hypothesis. Because IL-33 and IL-4 might have cooperative effects with IL-4 initiated and IL-33 amplified or vice versa, an *in vitro* experiment with various combinations of IL-4 and IL-33 to show the conversion of Tc into Tc-2 is needed to see if this supports the hypothesis.

Response: The reviewer raises an excellent point and we agree that the proposed experiment has the potential to add valuable insights into how IL-33 promotes Tc2 cell formation. We therefore isolated splenic Tc cells from wildtype mice and cultured the cells with various combinations of IL-4 and IL-33. These experiments revealed that GATA3, IL-5 and IL-13 levels were increased in Tc cells by the addition of IL-4, but not by IL-33 exposure alone (**Reviewer Fig.6**). Interestingly, co-treatment with IL-4 and IL-33 did not further enhance the type-2 skewing induced by IL-4 alone, despite robust induction of IL-33R/ST2 expression (**Reviewer Fig.6**). These results suggest that IL-33 acts in an indirect manner on Tc2 cell formation *in vivo*, possibly by increasing IL-4 levels in the tissue microenvironment - as was also suggested by the reviewer (see our response to comment #6 below for additional experimental evidence). This experiment has been included in the revised manuscript (**page 8, line 248-261; Fig.9**).

Reviewer Fig.6. (A) Representative flow cytometry protein level quantification of the transcription factors GATA3 and T-bet and the IL-33 receptor ST2 on cultured mouse Tc cells. **(B)** Quantification of IFN γ ⁺, IL-4⁺, IL-5⁺, IL-9⁺, IL-13⁺, and IL-5⁺IL-13⁺ cultured Tc cells measured by flow cytometry. Symbols represent individual mice, n=2 mice per group, bars indicate mean values.

5. The decreased Tc-2 cells in *Tnfaip3Lg*-KO mice might be due to the lower availability of type 2 cytokines induced by Th cells, as in Fig. S7B. Lower availability of IL-4 may lead to the less conversion of Tc cells into Tc-2 cells. Nevertheless, if the *Tnfaip3Lg*-KO mice have low IL-33 levels in the BALF, this would at least be consistent with the author's hypothesis.

Response: We agree with the reviewer that lower IL-4 availability could be a downstream effect of IFN γ in this context, effectively reducing Tc2 formation. In the past, we have tried extensively to measure IL-33 levels in BAL fluid obtained from our AAI models. Unfortunately, IL-33 has in our hands proven to be a difficult cytokine to measure due to its low concentrations in HDM-driven models, unknown kinetics and the complex combinations of active and inactive forms of the IL-33 molecule that exist *in vivo* (PMID: 35640416¹¹).

6. The study lacks mechanistic experiments to determine how IL-33 induces Tc-2 cell generation. The conclusion is drawn based on the preliminary observation of decreased Tc-2 cell numbers in IL-33 deficient mice. RNA-sea of IL-33 treated CD8⁺ T cells to identify factors responsible for Tc-1 to Tc-2 conversion and other mechanistic studies seem important.

Response: Our new *in vitro* experiments show that IL-33 most likely does not directly induce Tc2 cell formation (see response above to comment #4). IL-33 by itself did not show any capacity to induce Tc2

formation *in vitro*, and co-treatment of Tc cells with IL-4 and IL-33 did not further enhance the type-2 skewing induced by IL-4 alone (**Reviewer Fig.6**). Hence, a transcriptomic analysis of Tc cells exposed to IL-33 *in vitro* cannot inform on the mechanisms underlying IL-33-dependent Tc2 formation.

These data imply that the strong effect of IL-33 on Tc2 formation *in vivo* (as supported by **Fig.6-8**) cannot be explained by direct ST2-mediated mechanism. The most plausible alternative scenario is that IL-33 instead promotes type-2 skewing of Tc cells by altering the lung microenvironment, likely by promoting local IL-4 release that subsequently induced type-2 cytokine production in Tc cells by increasing GATA3 levels. To further substantiate this hypothesis, we exposed wildtype mice to IL-33 and measured intracellular IL-4 levels in a wide range of pulmonary immune cell types. This experiment revealed a substantial increase in IL-4⁺ Th cells, B cells, dendritic cells, basophils and eosinophils induced by IL-33 treatment (**Reviewer Fig.7**). In terms of abundance, the increase in IL-4⁺ eosinophils was particularly striking. Based on these observations, we believe that IL-33 evokes broad IL-4 production by various immune cells with key roles in eosinophilic airway inflammation, which in turn promotes type-2 skewing of Tc cells. This line of thinking and experimental work has been included in the revised manuscript (**page 8, line 248-261; page 10, line 310-315; Fig.9**), and our model has been adjusted accordingly (**Fig.10**).

Reviewer Fig.7. Numbers of IL-4 producing CD4⁺ and CD8⁺ T cells, B cells, group 2 innate lymphoid cells (ILC2s), natural killer (NK) cells, conventional DC (cDC) 1 and cDC2 cells, monocyte derived DCs (moDCs), plasmacytoid DCs (pDCs), basophils, mast cells and eosinophils determined in BAL of PBS or IL-33 treated mice by flow cytometry analysis. Symbols represent individual mice, n=6 mice per group, bars indicate mean values +/- SEM. *P<0.05, **P<0.01, ***P<0.001 (Mann-Whitney U test). BAL: bronchoalveolar lavage.

7. The presence of up to 60% IFN- γ ⁺ Th cells at a steady state in Fig. 4C is surprising. Mouse models have mainly naïve Th cells and do not produce IFN γ at steady state.

Response: We thank the reviewer for pointing this out, as we now realize that the scale in the previous Fig.4C was difficult to read for the PBS treated mice, making some values appear higher. As shown in **Reviewer Fig.8A**, we observed that ~25% of the Th cell population in PBS-treated mice has the capacity to produce IFN γ - we apologized for the confusion.

Important to note here is that naïve CD4⁺ T cells have the capacity to produce IFN γ upon PMA/ionomycin stimulation (see e.g. Han et al. PNAS 2012, PMID: 22160692¹²). For example, we and others have observed that human naïve CD4⁺ T cells receiving short-term (24h) TCR stimulation rapidly upregulate *IFNG* expression (**Reviewer Fig.8B-C**). Moreover, our control mice were not analyzed at 'steady-state', as they were exposed to a similar number of intratracheal injections (with PBS) as the HDM-treated animals, which may induce low levels of epithelial stress and T cell activation. To avoid any confusion and given that the PBS condition only serves as a control for absence of eosinophilic inflammation and T cell infiltration (as shown in revised **Fig.4B/D & Fig.4G/I**), we decided to only show the HDM and HDM+recIFN γ conditions for the population skewing analyses (revised **Fig.4C/H**).

Reviewer Fig.8. (A) Flow cytometry analysis of IFN γ expression in CD4⁺ T cells isolated from wildtype mice on an experimental HDM-driven allergic airway inflammation protocol (see Fig.4A of the revised manuscript). ‘PBS’ mice were treated intratracheally with PBS instead of HDM. Cytokine measurements were conducted after in vitro restimulation with PMA/Ionomycin. **(B)** mRNA levels of *IFNG* expressed by human naive CD4⁺ T cells before and after 24 hours of stimulation with IL-2 and anti-CD3/anti-CD28 beads (RNA-Seq data obtained from Calderon et al. ¹). **(C)** mRNA levels of *IFNG* expressed by human naive CD4⁺ T cells before and after 24 hours of stimulation with IL-2 and anti-CD3/anti-CD28 beads (unpublished RNA-Seq data).

8. In Fig. 4C, there is a large difference in IFN-g⁺ Th cells between the PBS challenged group (20-60%) vs mice challenged with HDM (1.5-3%). However, Fig.S6B shows similar cell numbers between these two groups. This must be resolved.

Response: We agree that absolute cell counts of IFN γ ⁺ Th cells are indeed similar between PBS and HDM treated mice (**Supplementary Fig.7C**), but are not sure why the reviewer finds this problematic. Upon HDM treatment, Th cell counts in the BAL increase dramatically (as expected, i.e. from 1*10⁴ to 7*10⁴ cells on average - see **Fig.4B**). Whereas a lower percentage of the pulmonary Th cells produces IFN γ after HDM exposure (1.5-3%), their absolute numbers may very well remain similar if influx or local expansion mostly involves IFN γ - cells. This is indeed the case, as we show in the pie charts in **Supplementary Fig.7D** and absolute cell count measurements in **Fig.4D** and **Supplementary Fig.7C**. To better clarify this notion we have moved panels showing absolute counts from supplementary to main figures and have added a sentence clarifying our use of percentages (i.e. to denote population skewing) (**page 6, line 164-168; Fig.4D/I**).

9. There is a huge variation in the percentage of IFN γ ⁺ Th cells and Tc cells with the PBS group in Figs. 4C and 6D.

Response: Differences in percentages of IFN γ ⁺ cells in PBS treated mice on a HDM experimental protocol (involving 5 intratracheal PBS injections) range from 20%-30% for Th cells (**Reviewer Fig.8**) and 35%-45% for Tc cells (**Fig.4**). In **Fig.6**, describing an IL-33 experimental protocol (involving 3 intratracheal PBS injections) we report percentages of IFN γ ⁺ cells ranging from 10%-18% for Th cells and 17%-24% for Tc cells. In our opinion, such differences in IFN γ production represent rather normal levels of intra-experimental variation.

References

1. Calderon, D. *et al.* Landscape of stimulation-responsive chromatin across diverse human immune cells. *Nat Genet* **51**, 1494-1505 (2019).
2. Guo, L. *et al.* Innate immunological function of TH2 cells in vivo. *Nat Immunol* **16**, 1051-1059 (2015).
3. Freeman, B.E., Hammarlund, E., Raue, H.P. & Slifka, M.K. Regulation of innate CD8⁺ T-cell activation mediated by cytokines. *Proc Natl Acad Sci U S A* **109**, 9971-9976 (2012).
4. Seumois, G. *et al.* Single-cell transcriptomic analysis of allergen-specific T cells in allergy and asthma. *Sci Immunol* **5** (2020).
5. Li, B.W. *et al.* T cells are necessary for ILC2 activation in house dust mite-induced allergic airway inflammation in mice. *Eur J Immunol* **46**, 1392-1403 (2016).
6. Klein Wolterink, R.G. *et al.* Pulmonary innate lymphoid cells are major producers of IL-5 and IL-13 in murine models of allergic asthma. *Eur J Immunol* **42**, 1106-1116 (2012).
7. de Kleer, I.M. *et al.* Perinatal Activation of the Interleukin-33 Pathway Promotes Type 2 Immunity in the Developing Lung. *Immunity* **45**, 1285-1298 (2016).

8. Tindemans, I. *et al.* Notch signaling in T cells is essential for allergic airway inflammation, but expression of the Notch ligands Jagged 1 and Jagged 2 on dendritic cells is dispensable. *J Allergy Clin Immunol* **140**, 1079-1089 (2017).
9. KleinJan, A. *et al.* The Notch pathway inhibitor stapled alpha-helical peptide derived from mastermind-like 1 (SAHM1) abrogates the hallmarks of allergic asthma. *J Allergy Clin Immunol* **142**, 76-85 e78 (2018).
10. Tindemans, I. *et al.* Notch signaling licenses allergic airway inflammation by promoting Th2 cell lymph node egress. *J Clin Invest* **130**, 3576-3591 (2020).
11. Cayrol, C. & Girard, J.P. Interleukin-33 (IL-33): A critical review of its biology and the mechanisms involved in its release as a potent extracellular cytokine. *Cytokine* **156**, 155891 (2022).
12. Han, Q. *et al.* Polyfunctional responses by human T cells result from sequential release of cytokines. *Proc Natl Acad Sci U S A* **109**, 1607-1612 (2012).

REVIEWER COMMENTS

Reviewer #1 (Remarks to the Author):

General comments:

The authors addressed most of my comments. For example, additional experiments have been performed to demonstrate the capacity of human Tc2 cells to produce type 2 cytokines and to examine the roles of IL-4 in explaining the effects of IL-33 to promote Tc2 cells. The results of these new experiments have been incorporated into the revised manuscript, and it has been clearly improved. However, one major comment remains.

Specific comments:

1. Previously, the presence of Tc2 cells in asthmatic patients has already been described (e.g., Cho SH, Am J Respir Crit Care Med 171:224-230, 2005, Hilvering B, Mucosal Immunol 12:581, 2019). This manuscript could provide novelty beyond the association between Tc2 cells and asthma. However, the results of the adoptive transfer experiments (new Figure 3) are predictable because the authors transferred Tc2 cells capable of producing type 2 cytokines. Thus, the important question remains whether and how much endogenous Tc2 cells (as compared to Th2 cells) play roles in airway inflammation during antigen-dependent or -independent type 2 immune responses. This reviewer would suggest deleting CD8+ T cells in the IL-33 model (Figure 6) and examine the impact of Tc2 depletion on eosinophilic inflammation and airway pathology.

Reviewer #2 (Remarks to the Author):

The authors have addressed my main comment. Several other new experiments further strengthen the manuscript.

Reviewer #3 (Remarks to the Author):

In this study, the authors provide evidence that IL-33 promotes type 2 cytokine production by lung Tc cells in allergic lung inflammation, with a link to asthma exacerbations. In the revised manuscript, the authors have done a good job of responding to the comments of all 3 reviewers, adding new data that enhance the study, for example showing that Tc2 cells can induce eosinophilic airway inflammation in vivo. Whereas the underlying mechanisms have not been fully elucidated, I believe this represents an important body of work and that it can be left to future studies to further explore the mechanisms.

Reviewer #1:*General comments:*

The authors addressed most of my comments. For example, additional experiments have been performed to demonstrate the capacity of human Tc2 cells to produce type 2 cytokines and to examine the roles of IL-4 in explaining the effects of IL-33 to promote Tc2 cells. The results of these new experiments have been incorporated into the revised manuscript, and it has been clearly improved. However, one major comment remains.

Response: We are glad to read that the reviewer concludes that new experiments have clearly improved our manuscript. Below we address the remaining comment.

Specific comments:

1. Previously, the presence of Tc2 cells in asthmatic patients has already been described (e.g., Cho SH, *Am J Respir Crit Care Med* 171:224-230, 2005, Hilvering B, *Mucosal Immunol* 12:581, 2019). This manuscript could provide novelty beyond the association between Tc2 cells and asthma. However, the results of the adoptive transfer experiments (new Figure 3) are predictable because the authors transferred Tc2 cells capable of producing type 2 cytokines. Thus, the important question remains whether and how much endogenous Tc2 cells (as compared to Th2 cells) play roles in airway inflammation during antigen-dependent or -independent type 2 immune responses. This reviewer would suggest deleting CD8+ T cells in the IL-33 model (Figure 6) and examine the impact of Tc2 depletion on eosinophilic inflammation and airway pathology.

Response: While the suggested experiment of examining the contribution of Tc2 cells to the inflammation in the IL-33 model is interesting, we and others have shown that even in a complete absence of T and B cells (i.e. Rag-deficient mice) the eosinophilia in this antigen-independent model can be fully maintained by ILC2s^{1,2}. In an antigen-dependent setting, a large body of experimental work using mouse models of allergic airway inflammation has already shown that removing CD8+ T cells - either through the use of knockout mice or via antibody-mediated depletion - can reduce airway hyperresponsiveness and eosinophilia³⁻⁸. For example, CD8+ T cell-deficient mice sensitized with house dust mite develop significantly lower airway hyperresponsiveness and have reduced eosinophil lung infiltration⁶. Finally, we would like to point out that our revised manuscript also contains adoptive transfer experiments with non-cultured naive Tc cells, which showed type-2 skewing in vivo upon TCR stimulation - in particular when combined with IL-33 treatment (**Fig.7**).

In light of these arguments, we would prefer not to engage in CD8 depletion experiments ourselves, as we feel that these would contribute little to the existing literature. Important to emphasize is that our study's main novelties go beyond showing that Tc2 cells contribute to eosinophilia: 1) we provide the first links between Tc2 formation and exacerbations in asthma patients, and 2) we define key microenvironmental signals that promote (i.e. IL-33/IL-4) or suppress (i.e. IFN γ) Tc2 formation. We have modified our Introduction to more specifically highlight existing literature that has already shown the importance of endogenous CD8+ T cells for allergic airway inflammation (**page 3, line 43-45**).

Reviewer #2:

The authors have addressed my main comment. Several other new experiments further strengthen the manuscript.

Response: We thank the reviewer for this positive assessment of our revisions.

Reviewer #3:

In this study, the authors provide evidence that IL-33 promotes type 2 cytokine production by lung Tc cells in allergic lung inflammation, with a link to asthma exacerbations. In the revised manuscript, the authors have done a good job of responding to the comments of all 3 reviewers, adding new data that enhance the study, for example showing that Tc2 cells can induce eosinophilic airway inflammation in vivo. Whereas the underlying mechanisms have not been fully elucidated, I believe this represents an important body of work and that it can be left to future studies to further explore the mechanisms.

Response: We appreciate these positive remarks regarding our efforts to revise the manuscript and the importance of our study.

References

1. Klein Wolterink, R.G. *et al.* Pulmonary innate lymphoid cells are major producers of IL-5 and IL-13 in murine models of allergic asthma. *Eur J Immunol* **42**, 1106-1116 (2012).
2. Kondo, Y. *et al.* Administration of IL-33 induces airway hyperresponsiveness and goblet cell hyperplasia in the lungs in the absence of adaptive immune system. *Int Immunol* **20**, 791-800 (2008).
3. Jia, Y. *et al.* Stepwise epigenetic and phenotypic alterations poise CD8⁺ T cells to mediate airway hyperresponsiveness and inflammation. *J Immunol* **190**, 4056-4065 (2013).
4. Miyahara, N. *et al.* Effector CD8⁺ T cells mediate inflammation and airway hyper-responsiveness. *Nat Med* **10**, 865-869 (2004).
5. Miyahara, N. *et al.* Contribution of antigen-primed CD8⁺ T cells to the development of airway hyperresponsiveness and inflammation is associated with IL-13. *J Immunol* **172**, 2549-2558 (2004).
6. Raemdonck, K. *et al.* CD4(+) and CD8(+) T cells play a central role in a HDM driven model of allergic asthma. *Respir Res* **17**, 45 (2016).
7. Schaller, M.A., Lundy, S.K., Huffnagle, G.B. & Lukacs, N.W. CD8⁺ T cell contributions to allergen induced pulmonary inflammation and airway hyperreactivity. *Eur J Immunol* **35**, 2061-2070 (2005).
8. Hamelmann, E. *et al.* Requirement for CD8⁺ T cells in the development of airway hyperresponsiveness in a murine model of airway sensitization. *J Exp Med* **183**, 1719-1729 (1996).

REVIEWERS' COMMENTS

Reviewer #1 (Remarks to the Author):

The authors addressed my comments in the rebuttal letter. While this reviewer wishes that the authors performed the CD8+ T cell depletion experiments to verify the biological significance of Tc2 cells in their hands, the authors' rationale not to perform such experiments is considered reasonable. The revision of the manuscript that states the gaps in our previous knowledge more clearly has made the manuscript stronger. I have no more comments to this manuscript.

Reviewer #1:

The authors addressed my comments in the rebuttal letter. While this reviewer wishes that the authors performed the CD8+ T cell depletion experiments to verify the biological significance of Tc2 cells in their hands, the authors' rationale not to perform such experiments is considered reasonable. The revision of the manuscript that states the gaps in our previous knowledge more clearly has made the manuscript stronger. I have no more comments to this manuscript.

Response: We are happy to read that the reviewer has considered our rationale not to perform the experiment as reasonable and that modifications to the text have improved the manuscript.